# Synthesis and structural analysis of dinucleotides containing 2′,3′-*trans*-bridged nucleic acids with *trans*-5,6- or 5,7-fused ring skeleton
Takashi Osawa [1], Ryota Nakanishi[1], Keito Uda[1], So Muramoto[1] & Satoshi Obika [1,2] ✉

Artificial nucleic acids in which the conformation of the sugar or phosphate backbone of the oligonucleotide is appropriately fixed can form stable duplexes. In this study, we designed dinucleotides containing 2′,3′-*trans*-bridged nucleic acids (2′,3′-*trans*-BNAs) based on the idea that the sugar conformation and torsions angles δ, ε, ζ, α, and β of the backbone can be controlled by a 5,6- or 5,7-membered *trans*-fused ring skeleton cyclized between the 2′- and 3′-positions of the sugar moiety. Given that the construction of *trans*-5,6-fused ring skeletons is synthetically challenging, the synthesis was optimized and a detailed structural analysis of these new bridged 2′,3′-*trans*-BNA systems was conducted. The 2′,3′-*trans*-BNAs could be synthesized from a commercially available D-glucose derivative with the key intramolecular gold-catalyzed cyclization reaction achieved using a cyclization precursor bearing an intramolecular hydroxy group and an internal alkyne. Structural analysis of the 2′,3′-*trans*-BNAs showed an N-type sugar conformation for all the derivatives, which is similar to that in RNA-duplex, and the ζ and α torsion angles for the 2′,3′-*trans*-BNAs were a characteristic feature of the compounds that differ from the corresponding angles of the natural duplexes.

As drug targets for conventional pharmaceuticals such as small-molecule drugs are being exhausted, oligonucleotides[1–5] are attracting attention as a novel drug discovery modality. Oligonucleotide therapeutics, which can hybridize with disease-causing intracellular RNAs and control their functions, offer a new therapeutic strategy for intractable diseases. Oligonucleotides are also being investigated for applications other than pharmaceuticals, such as probes to detect target RNA[6,7] and oligonucleotide-based nanotechnology[8,9]. In these technologies, it is crucial that the oligonucleotides have high in vivo stability and can form stable duplexes via Watson–Crick base pairing[10,11]. On the other hand, since natural oligonucleotides have insufficient in vivo stability and binding affinity to the complementary strand, chemically modified artificial nucleic acids with improved functions are necessary for practical use.

Chemical modifications of the phosphate backbone of oligonucleotides can be an effective way of improving their function. For example, oligonucleotides modified with phosphorothioate (PS), in which one oxygen atom in the phosphate bond is replaced with a sulfur atom, have increased

nuclease resistance and are widely used in oligonucleotide therapeutics[12]. On the other hand, PS-modified oligonucleotide therapeutics still have issues such as toxicity due to nonspecific protein binding[13]; therefore, the development of artificial nucleic acids modified with phosphate backbone moieties other than PS is essential for the practical application of oligonucleotides. In this context, oligonucleotides that are linked by chemical bonds without phosphorus atoms have been explored[14–19]. The absence of phosphate groups in these chemically modified oligonucleotides greatly improves their nuclease resistance[20]. Particularly, oligonucleotides in which the phosphodiester is replaced by a formamacetal linkage are known to destabilize DNA duplexes while stabilizing RNA duplexes and are expected to be useful for targeting RNA[21].

The strategy of immobilizing sugar moieties is often employed in the development of chemically modified nucleic acids to improve their duplex formation ability. Sugar puckering in nucleic acids occurs such that the N- and S-type conformations are in equilibrium in single-stranded DNA and RNA, whereas the sugar conformation in DNA duplexes is S-type and that

[1]Graduate School of Pharmaceutical Science, Osaka University, 1-6 Yamadaoka, Suita, Osaka, 565-0871, Japan. [2]Institute for Open and Transdisciplinary Research Initiatives, Osaka University, 1-3 Yamadaoka, Suita, Osaka, 565-0871, Japan. ✉e-mail: obika@phs.osaka-u.ac.jp

**Fig. 1 | Artificial nucleic acids with fixed conformations of the sugar moiety or phosphate backbone. A** 2′,4′-BNA/LNA (**B**) bicyclo and tricyclo DNA, (**C**) 2′,3′-*trans*-BNAs. These compounds are typical artificial nucleic acids with fixed sugar puckering and torsion angles of the phosphate backbone.

in RNA duplexes is N-type[22,23]. Our group and Prof. Wengel's group have independently developed 2′,4′-bridged nucleic acid (2′,4′-BNA)[24,25] or locked nucleic acid (LNA, Fig. 1A)[26,27], which has a locked N-type sugar and can form highly stable duplexes with target RNA. To date, artificial nucleotides with bicyclic scaffolds and appropriately fixed conformations of the phosphate backbone, such as constrained nucleic acids (CNAs)[28–37] and some dinucleotides[38–44] have been developed. Among these artificial nucleotides, CNAs form stable duplexes with complementary single strands. Additionally, bicyclo DNAs[45], and tricyclo DNAs[45–48], which have a fixed torsion angle γ, are known to stabilize the duplexes formed with RNA. Moreover, these bicyclo and tricyclo DNAs have a fixed torsion angle δ, which is related to sugar puckering; the torsion angle δ is also fixed for the 2′,4′-BNA/LNA described above. Based on these findings, the design and synthesis of artificial nucleic acids with precisely fixed conformations of both the sugar moiety and phosphate backbone have been examined with the aim of developing new artificial nucleic acid materials with superior target RNA-binding ability. For example, TriNA[49] and α-L-TriNA[50], with a tricyclic scaffold that properly constrains the torsion angle γ of the phosphate backbone and tightly fixes the sugar conformation to the N-type, afford greatly enhanced RNA hybridization with oligonucleotides. Our group recently developed spiro-DNAs[51,52] and 3′,4′-tpBNAs[53] and found

that they have target RNA-selective duplex-forming ability and extremely high nuclease resistance. Additionally, oxabicyclic nucleoside phosphonates, developed by Seth and Hanessian[54], can fix the sugar moiety conformation and the torsion angle ε by introducing a unique and rigid *trans*-5,6-fused ring skeleton.

Some DNAs and RNAs function by adopting a non-canonical three-dimensional structure that differs from the usual double-helix structure. Aptamers[55–57] and ribozymes[58–60], which are representative examples of such oligonucleotides, have characteristic functional structures. It has been noted that the torsion angles α to ζ in their phosphate backbones differ from those of general DNA and RNA duplexes, depending on the respective molecules[61–68]. Against this background, we designed 2′,3′-5,6-*trans*-BNAs (**A** and **B**, Fig. 1C) with a *trans*-fused ring skeleton in which a six-membered ring is introduced at the 2′- and 3′-carbons of the sugar moiety. The introduction of this characteristic fused ring skeleton into the sugar moiety not only constrains the sugar conformation to the N-type, but also tightly fixes the torsion angles δ, ε, ζ, and α of the backbone structure. It will be interesting to study how the conformational fixation of the phosphate backbone by the introduced *trans*-fused ring in 2′,3′-*trans*-BNAs affects the function of the oligonucleotides. Furthermore, although methods for constructing *cis*-5,6-fused ring skeletons in carbohydrates have been

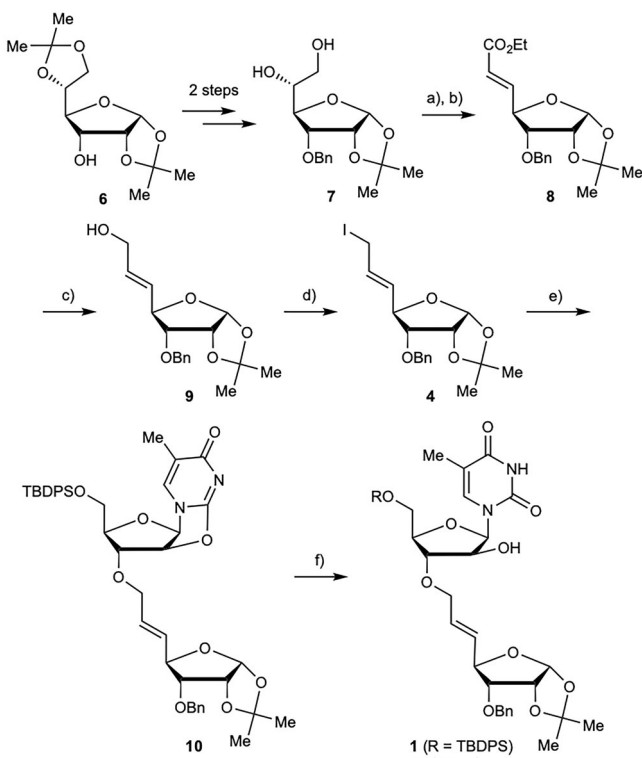

**Fig. 2 | Retrosynthetic analysis of 2′,3′-trans-BNAs for construction of trans-5,6- or 5,7-fused ring.** This scheme shows the substrates and starting materials for constructing trans-5,6- or 5,7-fused ring, which are the key reactions for synthesizing 2′,3′-trans-BNAs.

**Scheme 1 | Synthesis of dimer 1 with internal alkene.** Reagents and conditions: (**a**) NaIO₄, THF/H₂O (1:1), rt, 2 h; (**b**) Ph₃PCHCO₂Et, toluene, rt, 16 h, 73% (2 steps); (**c**) DIBAL, CH₂Cl₂, –78 °C, 2 h, 86%; (**d**) Ph₃P, I₂, imidazole, CH₂Cl₂, rt, 15 min, 76%; (**e**) see Table 1; (**f**) NaOH aq., EtOH, reflux, 20 h, 58% (**1**), 14% (**11**).

developed[69–72], the construction of trans-5,6-fused ring skeletons is still difficult. Therefore, the synthesis of 2′,3′-5,6-trans-BNAs will not only help elucidate how conformational restriction of nucleic acids can improve and control the function of oligonucleotides but also address a fascinating and challenging issue from the perspective of organic chemistry. From the two designed 2′,3′-5,6-trans-BNAs (**A** and **B**), we have achieved the synthesis of **B**. Surprisingly, 2′,3′-5,6-trans-BNA^olefin (**C**), which has a double bond inside the trans-fused ring, was found to be stable in these syntheses. In addition, the syntheses of 2′,3′-5,7-trans-BNAs (**D** and **E**, Fig. 1C) were performed to ascertain the structural properties of compounds with 5,7-trans-fused rings obtained in the desired fused ring construction. In this study, structural analysis of **B–E** was carried out by NMR measurements and computational methods, and the results were compared with the torsion angles of the phosphate backbone in natural DNA and RNA duplexes. This paper details the synthesis and structural analysis of 2′,3′-trans-BNAs **B–E**.

## Results and discussion
### Synthesis of 2′,3′-trans-BNAs
Considering the lack of reported methods to construct trans-5,6- or trans-5,7-fused ring skeletons, their synthesis was expected to be challenging. Therefore, to achieve the desired fused ring formation, dimers **1** and **2**, with both a 2′-hydroxy group and an internal olefin or internal alkyne, were designed as cyclization precursors (Fig. 2). We considered that the desired trans-fused ring skeleton could be constructed by intramolecular cyclization following activation of the internal olefin of dimer **1** by a Lewis acid, a Brønsted acid, or oxidative halogenating agents (Route A), or by intramolecular cyclization following activation of the internal alkyne of dimer **2** by a Lewis acid catalyst (Route B). Dimer precursors **1** and **2** can be obtained from allyl iodide **4** and propargyl iodide **5**, respectively.

First, dimer **1**, with an internal olefin, was synthesized according to Route A in Fig. 2 (Scheme 1). 1,2-Diol **7**[73,74] was prepared in a two-step process from the commercially available D-glucose derivative **6**. Subsequent periodate-mediated cleavage of the 1,2-diol was followed by conversion into

**Scheme 2 |** Coupling between allyl iodide 4 and compound 3.

**Table 1 | Optimization of conditions for coupling reaction between allyl iodide 4 and compound 3**

| Entry | Base | Solvent | Temp. (°C) | Yield (%) |
|---|---|---|---|---|
| 1 | NaH | DMF | 0 to rt | 20 (**12**) |
| 2 | NaHMDS | THF | –78 to 0 | complex mixture |
| 3 | NaH | DMF | –15 to 0 | 43 (**10**) |
| 4 | NaH | DMF | –15 | 69 (**10**) |

the unsaturated ester **8** via the Horner–Wadsworth–Emmons reaction. Allyl alcohol **9**, obtained by DIBAL-mediated reduction of compound **8**, was iodinated under the Appel reaction conditions to afford the desired allyl iodide **4**. Trials were then made to couple compound **4** with compound **3**[75] (Scheme 2 and Table 1). The reaction was first carried out at room temperature using NaH; however, this gave the undesired epoxide **12** (entry 1). This was attributed to the attack on the 2′-position by the 3′-oxygen anion generated by NaH treatment, resulting in ring-opening and N-allylation. Therefore, reaction conditions under which the epoxidation could be controlled were explored. When NaHMDS was used as an alternative base, the substrates decomposed, resulting in a complex mixture that was difficult to analyze (entry 2). When the temperature was lowered to 0 °C the reaction with NaH gave **10** in 43% yield (entry 3). Finally, when the reaction temperature was maintained at –15 °C, the yield of the desired product **10** was improved to 69% (entry 4). The resulting compound **10** was then hydrolyzed under basic conditions to give the desired compound **1** in 58% yield, along with a 14% yield of **11**, from which the TBDPS group was removed. The obtained cyclization precursor **1** was subjected to intramolecular cyclization using oxidative halogenating reagents (iodine and N-iodo-succinimide (NIS)), epoxidizing agents (mCPBA), or Lewis acids (Al(OTf)$_3$ and SnCl$_4$). However, the reactivity of the internal olefin of **1** was extremely low and the desired cyclization product was not obtained under any of the reaction conditions.

To construct the *trans*-fused ring via Route B (Fig. 2), the synthesis of cyclization precursor **2** was required (Scheme 3). Periodate-mediated cleavage of 1,2-diol **7** and treatment of the generated aldehyde with Ohira–Bestmann reagent gave alkyne **13**. Hydroxymethylation of the terminal alkyne was achieved using n-BuLi and paraformaldehyde to give compound **14**. In this reaction, gaseous formaldehyde was generated by heating paraformaldehyde, and the gas was then passed through the reaction solution[76,77]. Almost no byproducts were produced when gaseous formaldehyde was used, and product **14** was obtained in 85% yield. Compound **14** was then iodinated under the Appel reaction conditions to give the desired propargyl iodide **5**. Propargylation of compound **3** using compound **5** was then performed. Based on the findings shown in Table 1, compound **15** was obtained in 77% yield by maintaining the reaction temperature at –15 °C. The resulting compound **15** was then hydrolyzed to afford the desired precursor **2** in 72% yield, along with a 20% yield of compound **16**.

Intramolecular cyclization of the obtained compound **2** was attempted using a gold catalyst (Table 2). Based on the few reported syntheses of *trans*-5,6-fused rings[78], [(Ph$_3$PAu)$_3$O]BF$_4$ was selected as the gold catalyst. In the previous report, the reaction was carried out under reflux conditions using THF; however, the reaction did not proceed with substrate **2** under these conditions (entry 1). When the THF solvent was replaced with 1,4-dioxane

as a similar cyclic ether, the desired *trans*-fused ring skeleton was formed and cyclized products **17** and **18** were obtained as a mixture in 56% yield (entry 2). The resulting compounds **17** and **18** were isolated by silica gel column chromatography using a mixture of chloroform, dichloromethane, hexane, and acetonitrile (1:1:1:1) as eluents. From the COSY spectral analysis of the isolated compounds **17** and **18**, it was determined that compound **17** has a *trans*-5,6-fused ring, and compound **18** has a *trans*-5,7-fused ring (see the Supporting Information). Although the stereochemistry of the olefins in **17** and **18** produced by intramolecular cyclization could not be determined by NMR analysis, both olefins are likely to be in the *cis*-configuration because the *trans*-orientation of the two alkyl groups attached to the alkyne has been proposed as the reaction mechanism in the Au(I)-catalyzed hydroalkoxylation of alkynes[79]. In addition, the formation ratios of **17** and **18** were 1:2, based on NMR analysis (entry 2), indicating that 7-*endo*-cyclization was preferred under the reaction conditions. When the reaction was performed under reflux conditions with toluene, the seven-membered ring form **18** was generated almost exclusively, although the isolated yield was moderate (entry 3). To examine the selectivity of the 6-*exo*- and 7-*endo*-cyclization further, the reaction was performed in toluene at 80 °C; under these conditions, the ratio of **17**/**18** formed in the reaction was reduced to 1:7 (entry 4). This suggests that both the reaction temperature and solvent affect the selectivity of the reaction towards the formation of the seven-membered ring. Under reflux conditions using 1,2-dichloroethane (DCE) as the solvent, the total yield of the cyclized products improved to 82% and the ratio of **17** to **18** was 1:1, although a reaction time of 21 h was required (entry 5). Intramolecular cyclization was also performed using a combination of AgOTf and either Me$_2$SAuCl (entry 6) or Ph$_3$PAuCl (entry 7), which are monovalent gold catalysts similar to [(Ph$_3$PAu)$_3$O]BF$_4$. However, in both cases, the reaction afforded a complex mixture with no desired cyclization products. From these results, we conclude that the conditions given in entry 5 are optimal in terms of product yield and cyclization selectivity, although the reaction generates a mixture of 6-*exo*- and 7-*endo*-cyclization products **17** and **18**, respectively.

Compound **17**, with a *trans*-5,6-fused ring, was subjected to the contact reduction of intramolecular olefins (Scheme 4 and Table 3). Pd/C poisoned with ethylenediamine was used to prevent reductive deprotection of the Bn group. Surprisingly, under these conditions, the exocyclic olefin migrated into the fused ring to give compound **19** in 49% yield (entry 1). Therefore, the simultaneous reduction of the olefin and deprotection of the Bn group was attempted using a combination of 20% Pd(OH)$_2$/C and hydrogen. However, compound **20**, in which the Bn group was removed and the olefin was rearranged, was obtained in 85% yield (entry 2). Increasing the hydrogen pressure was ineffective in reducing the internal olefin in **19** (entry 3), resulting in the generation of only **20**. Furthermore, when cyclohexene was used as the hydrogen source for the catalytic reduction of the olefin in **20**, a complex mixture was formed (entry 4). The hydrogenation of **17** and **20** using Crabtree's catalyst, which has been used for the hydrogenation of tetrasubstituted olefins[80], was also investigated (entries 5 and 6); however, no reaction was observed. Thus, it was clear that the reduction of the rearranged olefin in the fused ring was difficult. The optimum conditions given in entry 2 were therefore used to obtain compound **20**, which was then used for the synthesis of 2′,3′-5,6-*trans*-BNAs.

The acetonide group was removed from **20** under acetic acidic conditions, followed by acetylation of the hydroxy group and Vorbrüggen

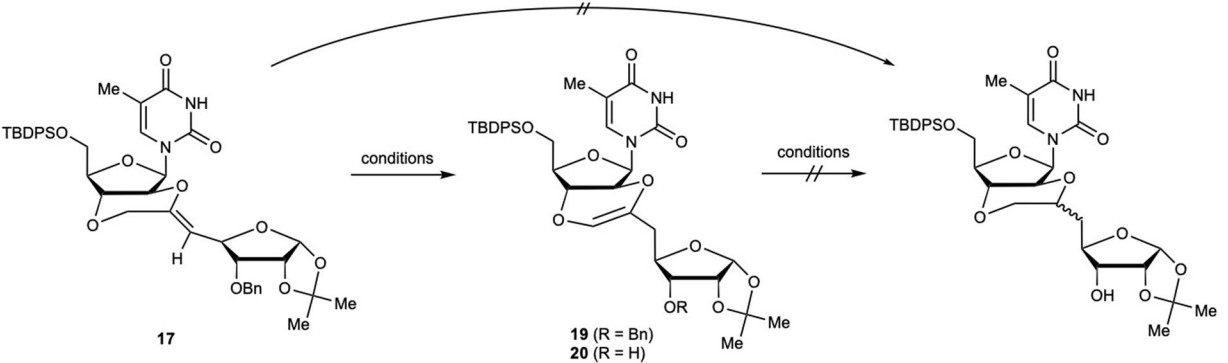

**Scheme 3 |** Synthesis of dimer **2** with internal alkyne and construction of *trans*-fused ring. Reagents and conditions: (**a**) NaIO$_4$, THF/H$_2$O (1:1), rt, 1 h; (**b**) Ohira–Bestmann reagent, K$_2$CO$_3$, MeOH, rt, 14 h, 82% (2 steps); (**c**) *n*-BuLi, (HCHO)$_n$, THF, −78 °C to rt, 1.5 h, 85%; (**d**) Ph$_3$P, I$_2$, imidazole, CH$_2$Cl$_2$, rt, 15 min, 91%; (**e**) compound **3**, NaH, DMF, −15 °C, 4 h, 77%; (**f**) NaOH aq., EtOH, reflux, 21 h, 72% (**2**), 20% (**16**) (**g**) see Table 2.

**Table 2 | Gold-catalyzed intramolecular cyclization of compound 2**

| Entry | Catalyst (5 mol%) | Solvent | Temp. (°C) | Time (h) | Ratio (17/18) | Yield (17 + 18) (%) |
|---|---|---|---|---|---|---|
| 1 | [(Ph$_3$PAu)$_3$O]BF$_4$ | THF | reflux | 3 | – | no reaction |
| 2 | [(Ph$_3$PAu)$_3$O]BF$_4$ | 1,4-dioxane | reflux | 2 | 1:2 | 56 |
| 3 | [(Ph$_3$PAu)$_3$O]BF$_4$ | toluene | reflux | 1 | <1:20 | 38 |
| 4 | [(Ph$_3$PAu)$_3$O]BF$_4$ | toluene | 80 | 14 | 1:7 | 23 |
| 5 | [(Ph$_3$PAu)$_3$O]BF$_4$ | DCE | reflux | 21 | 1:1 | 82 |
| 6 | Me$_2$SAuCl, AgOTf | DCE | reflux | 5 | – | complex mixture |
| 7 | Ph$_3$PAuCl, AgOTf | DCE | reflux | 5 | – | complex mixture |

**Scheme 4 |** Reductive hydrogenation of an exocyclic olefin in compound 17.

glycosylation[81,82] to give dimer **21** in ~40% yield (Scheme 5). The resulting compound **21** was treated with methanolic NaOMe solution to remove the acetyl group and afford the desired compound **22** in good yield (81%). Finally, the TBDPS group was removed from compound **22** by treatment with TBAF to obtain 2′,3′-5,6-*trans*-BNA$^{olefin}$ **23**, which was followed by reductive hydrogenation of the internal olefin of the resulting compound **23** to obtain 2′,3′-5,6-*trans*-BNA **24**. In this hydrogenation of 2′,3′-5,6-*trans*-BNA$^{olefin}$ **23**, 2′,3′-5,6-*trans*-BNA **24** was obtained as a single isomer, the

**Table 3 | Reductive hydrogenation of an exocyclic olefin in compound 17**

| Entry | Starting material | Reagent | H₂ source | Solvent | Temp. | Time (h) | Yield |
|---|---|---|---|---|---|---|---|
| 1 | 17 | Pd/C (en) | H₂ gas (1 atm) | THF | rt | 24 | 49% (19) |
| 2 | 17 | 20% Pd(OH)₂/C | H₂ gas (1 atm) | THF | rt | 24 | 85% (20) |
| 3 | 17 | 20% Pd(OH)₂/C | H₂ gas (5 atm) | THF | rt | 24 | 39% (20) |
| 4 | 20 | 20% Pd(OH)₂/C | cyclohexene | THF | reflux | 24 | Complex mixture |
| 5 | 20 | Crabtree's catalyst | H₂ gas (1 atm) | CH₂Cl₂ | rt | 24 | No reaction |
| 6 | 17 | Crabtree's catalyst | H₂ gas (5 atm) | CH₂Cl₂ | rt | 24 | No reaction |

**Scheme 5 |** Synthesis of 2′,3′-5,6-*trans*-BNA^olefin 23 and 5,6-*trans*-BNA 24. Reagents and conditions: (**a**) 60% AcOH, reflux, 2 h; (**b**) Ac₂O, Et₃N, DMAP, CH₂Cl₂, rt, 1 h; (**c**) thymine, BSA, TMSOTf, DCE, reflux, 14 h, 39% (3 steps from 20); (**d**) NaOMe, MeOH, rt, 22 h, 81%; (**e**) TBAF, THF, rt, 5 h, 76%; f) H₂ (1 atm), Pd(OH)₂/C, THF/MeOH (3:1), rt, 23 h, 53%.

**Scheme 6 |** Synthesis of 2′,3′-5,7-*trans*-BNAs 28 and 29. Reagents and conditions: (**a**) Ac₂O, H₂SO₄, AcOH, rt, 2 h; (**b**) thymine, BSA, TMSOTf, DCE, reflux, 14 h, 37% (two steps from 18); (**c**) K₂CO₃, MeOH, rt, 5 h, 95%; (**d**) TBAF, THF, rt, 6 h, 85%; (**e**) H₂ (1 atm), Pd(OH)₂/C, THF, rt, 23 h, 47% (28/29 = 3:7).

stereoconfiguration of which was determined by ¹H NMR and H-H COSY analysis. The coupling constants (*J* = 3.0, 12.0 Hz) of 24 indicated that the hydrogen of the chiral C–H produced by the hydrogenation reaction adopted a 1,2-diaxial relationship with the hydrogen on the six-membered ring. Thus, the isolated compound 24 was determined to be the *S*-isomer with a six-membered ring with the 5′-carbon in the equatorial position. Since the olefin inside the fused ring of compound 23 could be hydrogenated, glycosylation of 17 was attempted to obtain the *R*-isomer of 2′,3′-5,6-*trans*-BNA 24. However, the glycosylation was not successful, probably due to the poor stability of the exocyclic olefin.

The synthesis of 2′,3′-5,7-*trans*-BNA was carried out using compound 18 (Scheme 6). Deprotection of the acetonide group and acetylation with acetic anhydride and catalytic amount of sulfuric acid, followed by Vorbrüggen glycosylation to introduce the thymine unit, afforded compound 25 in 37% yield. The acetyl group was removed by treatment with potassium carbonate in methanol, and the TBDPS group was removed using TBAF to give compound 27. Reductive deprotection of the benzyl group and hydrogenation of the internal olefin using 20% Pd(OH)₂/C afforded 2′,3′-5,7-*trans*-BNAs 28 and 29 in 47% yield as a diastereomeric mixture with a ¹H NMR ratio of ~3:7. This mixture was purified by HPLC to isolate compounds 28 and 29 for NMR analysis. The stereoconfiguration of the chiral carbons produced by hydrogenation was determined by ¹H, H-H COSY, and NOESY NMR spectroscopic measurements and by computational conformational analysis (see below for computational details).

NOESY analysis revealed characteristic cross-peaks for compounds 28 and 29 (Fig. 3). The computational analyses of the *R*- and *S*-isomers were consistent with the observed NOE cross-peak correlations for the proximal protons in the 3D structure. Furthermore, ¹H NMR analyses of compounds 28 and 29 showed that the hydrogen atoms on the tertiary carbon produced by hydrogenation adopt a diaxial relationship with one of the hydrogen atoms of the CH₂ group in the seven-membered ring, which is consistent with the stable structures shown in Fig. 3. From these results, compounds 28 and 29 were determined to be the *R*- and *S*-isomer, respectively.

**Structural analysis of 2′,3′-*trans*-BNAs**
For the 2′,3′-*trans*-BNAs (compounds 23, 24, 28, and 29), the sugar conformation was determined by ¹H NMR spectroscopic analysis (Fig. 4). The coupling constants were calculated for the protons at the 2′-, 3′-, and 4′-positions to analyze the conformation. The signal from the 3′-hydrogen of all the compounds was a triplet that was coupled to the 2′- and 4′-protons with *J* values of 8.0–9.5 Hz, suggesting that all the protons in positions 2' to 4' adopt pseudo-axial conformations. This confirms that the sugar moiety of the 2′,3′-*trans*-BNAs is constrained to the N-type conformation, as determined by the *trans*-5,6- or *trans*-5,7-fused ring skeleton.

The dihedral angles of the backbone moiety of these four artificial nucleosides were then calculated using computational methods and compared with those of the natural A-type and B-type DNA duplexes[83]. In this conformational analysis, the model compounds 23a, 24a, 28a, and 29a, in which the 3′-side of the nucleoside was replaced with a methyl group, were used (Fig. 5). The torsion angles δ, ε, ζ, and α for 2′,3′-5,6-*trans*-BNAs 23a and 24a, and δ, ε, ζ, α, and β for 2′,3′-5,7-*trans*-BNAs 28a and 29a are

**Fig. 3 | Determination of stereo configuration of chiral carbons on 7-membered ring in compounds 28 and 29.** The most stable structures of compounds 28 (*R*-isomer) and 29 (*S*-isomer) shown below this figure analyzed by computational methods were consistent with the observed NOE cross-peak correlations for the proximal protons in the 3D structure.

**Fig. 4 | Conformational analysis of the sugar moiety of 2′,3′-*trans*-BNAs (23, 24, 28, and 29) by ¹H NMR measurement.** The results of the NMR spectroscopic analysis of the compounds 23, 24, 28, and 29 indicated that the sugar puckering of all 2′,3′-*trans*-BNAs is the N-type.

summarized in Table 4. The values of δ are similar to those of natural RNA duplexes with an A-type helix for all 2′,3′-*trans*-BNAs. These results strongly suggest that the sugar conformation of 2′,3′-*trans*-BNAs is the N-type, which is consistent with the results shown in Fig. 4. The dihedral angles ε and β of 2′,3′-*trans*-BNAs are in the antiperiplanar range and do not differ significantly from that of natural DNA or RNA duplexes. On the other hand, the dihedral angle ζ is similar to that of RNA for compounds 24a and 28a but significantly different for compounds 23a and 29a, which is possibly due to the introduction of a double bond inside the fused ring in 2′,3′-5,6-*trans*-BNA^olefin 23a and the fluctuation of the seven-membered ring structure in the *S*-isomer of 2′,3′-5,7-*trans*-BNA 29a. Furthermore, the values of the torsion angle α for compound 29a are close to those of RNA, while torsion

angle α for the other three compounds (23a, 24a, and 28a) are each characteristic angles that differ significantly from those in natural duplexes. From these results, the 2′,3′-5,6-*trans*-BNAs and 2′,3′-5,7-*trans*-BNAs may have dihedral angles ζ or α fixed in a characteristic conformation that differ from those of natural DNA and RNA, probably due to the rigid *trans*-fused ring skeleton of 2′,3′-*trans*-BNAs. Conversely, the *trans*-fused rings in 2′,3′-*trans*-BNAs are replaced by a ribonucleoside that is much larger than the methyl group, and the possibility that the conformation of the fused ring skeleton of the dimer may change under the influence of the stacking interaction of the two thymine nucleobases must be considered. Therefore, CD spectra were measured for the four dimers (23, 24, 28, and 29), and the obtained spectra were compared with those of typical DNA and RNA

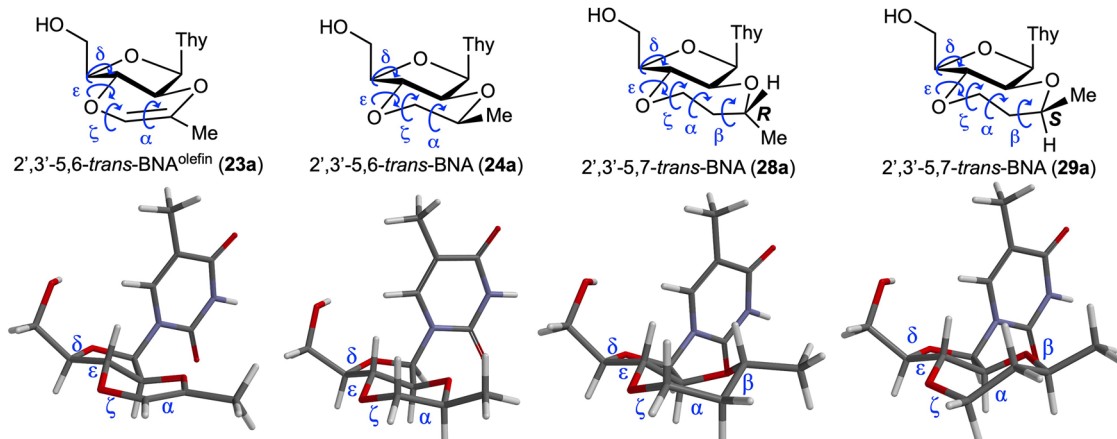

**Fig. 5 | Computational structural analysis of model 2′,3′-*trans*-BNAs 23a, 24a, 28a, and 29a (theoretical calculations were carried out at the DFT ωB97XD/6-31 G* level).** This figure shows the most stable structures of each of the model compounds **23a**, **24a**, **28a**, and **29a**, and the torsion angles of the phosphate backbone in the obtained structures are summarized in Table 4.

**Table 4 | Torsion angles of the backbone in the structure of 2′,3′-*trans*-BNAs shown in Fig. 5**

| | Torsion angle (°) | | | | |
|---|---|---|---|---|---|
| | δ | ε | ζ | α | β |
| RNA duplex (A type)[83] | 81 ± 7 | −157 ± 12 | −71 ± 12 | −67 ± 17 | 174 ± 14 |
| DNA duplex (B type)[83] | 128 ± 13 | −176 ± 11 | −95 ± 10 | −62 ± 15 | 176 ± 9 |
| 2′,3′-5,6-*trans*-BNA^olefin (**23a**) | 85 | 163 | −16 | 176 | – |
| 2′,3′-5,6-*trans*-BNA (**24a**) | 83 | 175 | −55 | 172 | – |
| 2′,3′-5,7-*trans*-BNA (**28a**, *R*-isomer) | 85 | −169 | −59 | 69 | 159 |
| 2′,3′-5,7-*trans*-BNA (**29a**, *S*-isomer) | 83 | 158 | 29 | −79 | −170 |

dimers (dT- and rT-dimers) and LNA-T-dimers with fixed sugar puckering (Fig. 6). The CD spectra of the four dimers were not identical to those of dT-, rT-, and LNA-T-dimers. While these results, in isolation, do not suggest that the two thymine nucleobases in 2′,3′-*trans*-BNAs (**23**, **24**, **28**, and **29**) do not have any stacking interaction, the structure of our dimer would not be that of a typical oligonucleotide helix. Determination of whether the introduction of a nucleoside into the *trans*-fused ring changes the orientation of the 6- or 7-membered rings is difficult based on the results of CD spectral measurements and conformational calculations. Although this structural analysis compared the structures of 2′,3′-*trans*-BNAs and a typical double helix, ribozymes and aptamers often contain bulge or stem-loop structures, which can have characteristic conformations that differ from the usual double helix conformations. Therefore, it is of great interest from the viewpoint of nucleic acid chemistry to study how 2′,3′-*trans*-BNAs, with their characteristic structure, affect the functions of oligonucleotides. For instance, since an aptamer has been shown by molecular dynamics simulations to contain nucleotides with torsion angles α and ζ that are within the antiperiplanar and synclinal ranges, respectively[61,66], it will be interesting to investigate how its binding activity changes by incorporating 2′,3′-5,6-*trans*-BNA **24** into this aptamer. In the future, we would like to introduce the synthesized 2′,3′-*trans*-BNAs into oligonucleotides based on the phosphoramidite chemistry[84–86] to address this question, despite the challenges of introducing appropriate protective groups selectively to the 2′-OH group in 2′,3′-*trans*-BNAs.

## Conclusion

We designed and synthesized dinucleotides containing 2′,3′-*trans*-BNAs as new artificial nucleic acid materials to investigate the effect of fixing the conformation of the sugar moiety and phosphate backbone of oligonucleotides on their function. The sugar moieties of the 2′,3′-*trans*-BNAs are constrained to the N-type conformation and the backbone torsion angles δ, ε, ζ, α and β are fixed by a rigid *trans*-fused ring skeleton bridged at the 2′- and 3′-positions of the sugar moiety. The *trans*-5,6- or *trans*-5,7-fused ring skeleton was constructed by a gold-catalyzed intramolecular cyclization of a precursor featuring both a 2′-hydroxy group and an internal olefin or internal alkyne, and the syntheses of 2′,3′-5,6-*trans*-BNA^olefin, 2′,3′-5,6-*trans*-BNA, and 2′,3′-5,7-*trans*-BNA were achieved. Structural analysis of the synthesized 2′,3′-*trans*-BNAs showed that the sugar conformation of all the 2′,3′-*trans*-BNAs is tightly constrained in the N-type, as expected. Moreover, the computational structural analysis revealed that whereas the δ and ε torsion angles of the 2′,3′-*trans*-BNAs are similar to those of the natural RNA duplex, the torsion angles ζ and α present a characteristic conformation that differs from those of the natural DNA and RNA duplexes. In addition, upon comparison of the 2′,3′-*trans*-BNAs developed in this study and previously synthesized nucleotides devoted to mimic local nucleic acid sugar/phosphate backbone distortion[87], it was evident that the structures were not identical. Therefore, 2′,3′-*trans*-BNAs appear to be a structurally unique nucleic acid analog. By contrast, 2′,3′-5,7-*trans*-BNA derivatives with an internal olefin in the 7-membered ring (2′,3′-5,7-*trans*-BNA^olefin) were not obtained when removing the benzyl group of compound **27**. If 2′,3′-5,7-*trans*-BNA^olefin can be obtained by replacing the benzyl group with an appropriate protecting group, a structural analysis should be performed. In summary, this study not only achieved the synthesis of artificial nucleic acids with a *trans*-5,6- or 5,7-fused ring skeleton, which is challenging from the viewpoint of synthetic organic chemistry but also developed attractive molecules from the perspective of nucleic acid chemistry. The influence of the unique structure of the 2′,3′-*trans*-BNAs on the properties of the oligonucleotides will be clarified in the future.

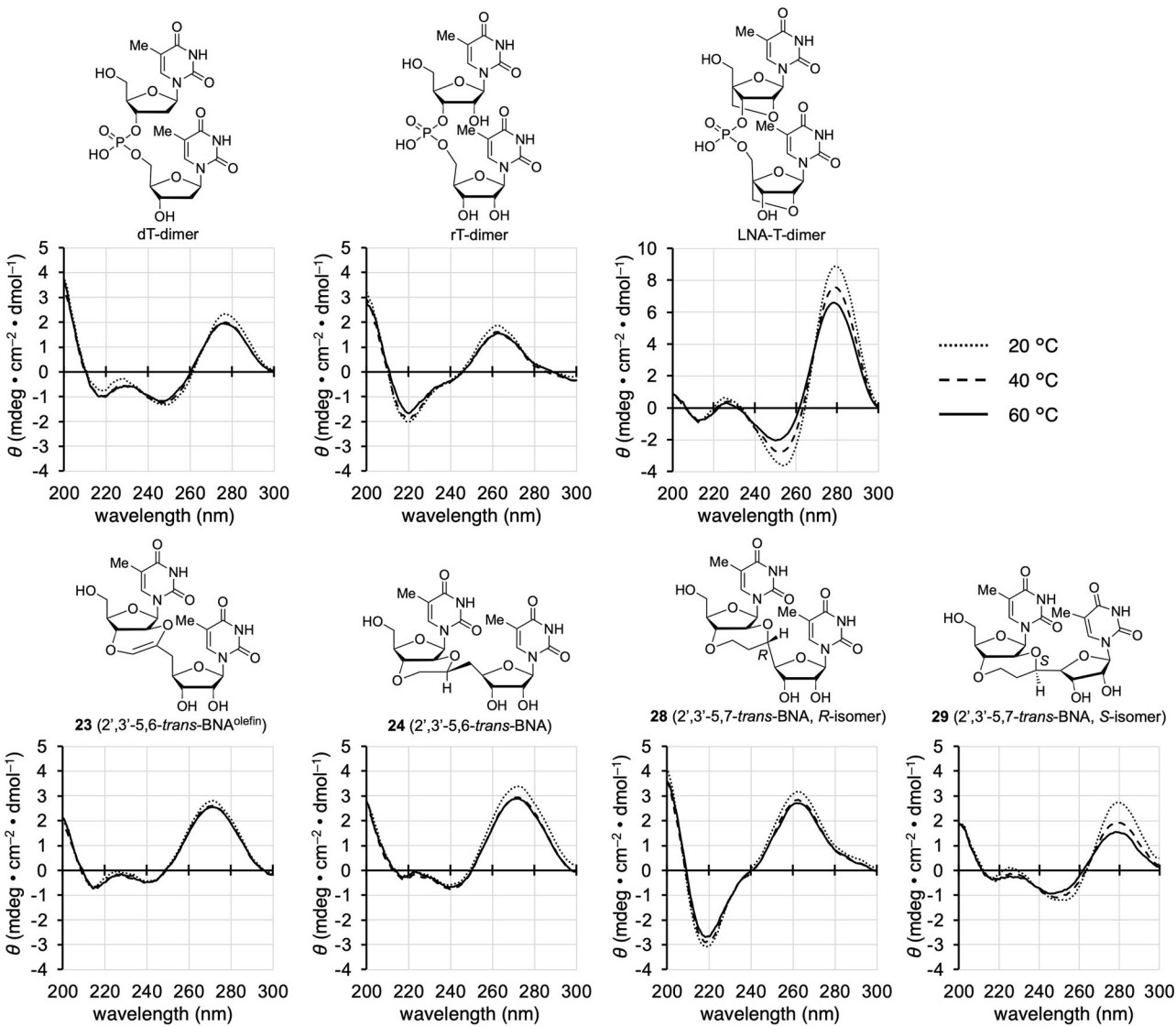

**Fig. 6 | CD spectra of dimers (dT-dimer, rT-dimer, LNA-T-dimer, 23, 24, 28, and 29).** Conditions: 10 mM sodium phosphate (pH 7.0) and 100 μM dimer in H$_2$O and MeOH (1:1) at 20, 40, and 60 °C. Comparing the CD spectra of the seven dimers,

the structure of 2′,3′-*trans*-BNAs appears to be different from that of typical nucleotide dimers (dT-dimer, rT-dimer, LNA-T-dimer).

## Methods
### General

All moisture-sensitive reactions were conducted in well-dried glassware under N$_2$ atmosphere. The progress of the reaction was monitored by analytical thin-layer chromatography (TLC) on pre-coated glass sheets (Silica gel 60 F254 by Merck). For column chromatography, silica gel PSQ-100B (Fuji Silysia Chemical Ltd.) were used. For celite filtration, Celite® 545RVS (Nacalai Tesque, Inc.) was used. UCR-150N-S (Techno-Sigma, Inc.) was used as a low-temperature reactor. CH-200 (Ishii Laboratory Works Co., Ltd) was used as a medium pressure hydrogenator. IR spectra were recorded on a JASCO FT/IR-4200 spectrometer. NMR experiments were performed on JEOL JNM-AL300, JNM-ECS400, JNM-ECS500. $^1$H NMR spectra were recorded at 300, 400, and 500 MHz. $^{13}$C NMR spectra were recorded at 100 and 126 MHz. Chemical shifts (δ) are expressed in ppm relative to internal tetramethylsilane (δ = 0.00 ppm), residual CDCl$_3$ (δ = 7.26 ppm), CD$_3$CN (δ = 1.94 ppm), CD$_3$OD (δ = 3.31 ppm) or DMSO-$d_6$ (δ = 2.50 ppm) for $^1$H NMR spectra, and residual CDCl$_3$ (δ = 77.0 ppm), CD$_3$OD (δ = 49.0 ppm) or CD$_3$CN (δ = 1.3 ppm) for $^{13}$C NMR spectra. MALDI-TOF mass spectra of new compounds were recorded on JEOL SpiralTOF JMS-S3000. For high performance liquid chromatography

(HPLC), SHIMADZU LC-20AT, CTO-20A, CBM-20A, SPD-20A and FRC-10A were used.

### Ethyl (*E*)-3-[(4 *R*)-3-*O*-benzyl-1,2-isopropylidene-α-D-erythrofur-anos-4-yl]-2-propenoate (8)

NaIO$_4$ (8.6 g, 40 mmol) was added to a solution of compound 7[73,74] (9.6 g, 31 mmol) in H$_2$O (50 mL) and THF (50 mL) at room temperature. The reaction mixture was stirred for 2 h at room temperature. The reaction mixture was filtered by vacuum filtration, and the filtrate was extracted with AcOEt. The organic phase was washed with water and brine, dried over Na$_2$SO$_4$, and concentrated *in vacuo*. The obtained residue (7.8 g) was dissolved in anhydrous toluene (100 mL) under N$_2$ atmosphere at room temperature. Ph$_3$PCHCO$_2$Et (14 g, 40 mmol) was added to the solution at room temperature. The reaction mixture was stirred for 16 h at room temperature. The resulting mixture was concentrated *in vacuo* and the residue (22 g) was purified by column chromatography (*n*-hexane/AcOEt = 5:1) two times to give compound 8 (7.7 g, 73%, two steps from compound 7) as a colorless oil. IR ν$_{max}$ (KBr): 2986, 2936, 1719, 1664, 1455, 1373, 1308, 1265 cm$^{-1}$. $^1$H-NMR (500 MHz, CDCl$_3$): δ = 7.36–7.29 (m, 5H), 6.91 (dd, *J* = 15.7, 5.1 Hz, 1H), 6.13 (d, *J* = 15.7 Hz, 1H), 5.77 (d, *J* = 3.7 Hz, 1H), 4.74

(d, $J$ = 12.0 Hz, 1H), 4.65–4.56 (m, 3H), 4.21 (q, $J$ = 7.1 Hz, 2H), 3.53 (dd, $J$ = 9.1, 4.2 Hz, 1H), 1.37 (s, 3H), 1.61 (s, 3H), 1.30 (t, $J$ = 7.1 Hz, 3H),. $^{13}$C NMR (100 MHz, CDCl$_3$): $\delta$ = 166.0, 143.5, 137.1, 128.5, 128.2, 128.1, 122.5, 113.3, 104.0, 81.9, 77.5, 72.5, 60.5, 26.8, 26.5, 14. HRMS (MALDI): calcd for C$_{19}$H$_{24}$O$_6$Na [MNa$^+$] 371.1471, found 371.1465.

### (*E*)-3-[(4 *R*)-3-*O*-Benzyl-1,2-isopropylidene-α-ᴅ-erythrofuranos-4-yl]-2-propen-1-ol (9)

Under N$_2$ atmosphere, DIBAL (1.0 M in *n*-hexane, 66 mL, 66 mmol) was added dropwise to a solution of compound **8** (7.7 g, 22 mmol) in anhydrous CH$_2$Cl$_2$ (100 mL) at −78 °C. The reaction mixture was stirred for 2 h at −78 °C. After completion of the reaction, excess AcOEt was added to the solution and then the reaction mixture was quenched with MeOH at −78 °C. Then, excess 30 (w/v)% Rochelle salt solution was added to the solution and the reaction mixture was further stirred for 30 min at room temperature. The whole mixture was extracted with CH$_2$Cl$_2$ and the organic phase was washed with 30 (w/v)% Rochelle salt solution and brine, dried over Na$_2$SO$_4$. The resulting mixture was concentrated *in vacuo* and the residue (6.5 g) was purified by column chromatography (*n*-hexane/AcOEt = 3:2) to give compound **9** (5.8 g, 86%) as a colorless oil. IR $\nu_{max}$ (KBr): 3391, 2987, 2934, 2871, 1455, 1374, 1251 cm$^{-1}$. $^1$H-NMR (500 MHz, CDCl$_3$): $\delta$ = 7.36–7.29 (m, 5H), 6.02 (dtd, $J$ = 15.6, 5.1, 0.9 Hz, 1H), 5.74 (d, $J$ = 3.7 Hz, 1H), 5.64 (dtd, $J$ = 15.5, 6.9, 1.6 Hz, 1H), 4.76 (d, $J$ = 12.2 Hz, 1H), 4.60–4.57 (m, 2H), 4.15 (t, $J$ = 5.5 Hz, 2H), 4.51–4.48 (m, 1H), 3.51 (dd, $J$ = 9.0, 4.3 Hz, 1H), 1.62 (s, 3H), 1.36 (s, 3H), 1.30 (t, $J$ = 6.1 Hz, 1H). $^{13}$C-NMR (100 MHz, CDCl$_3$): $\delta$ = 137.5, 134.0, 128.4, 128.1, 128.0, 127.3, 113.0, 103.7, 81.9, 78.1, 77.5, 72.4, 62.8, 26.7, 26.4. HRMS (MALDI): calcd for C$_{17}$H$_{22}$O$_5$Na [MNa$^+$] 329.1365, found 329.1359.

### (4 *R*)-(*E*)-3-*O*-Benzyl-4-(3-iodoprop-1-en-1-yl)-1,2-iso-propylidene-α-ᴅ-erythrofuranose (4)

Under N$_2$ atmosphere, I$_2$ (629 mg, 2.48 mmol) was added to a solution of Ph$_3$P (650 mg, 2.48 mmol) and imidazole (169 mg, 2.48 mmol) in anhydrous CH$_2$Cl$_2$ (15 mL) at room temperature. After the reaction mixture was stirred for 15 min at room temperature, a solution of compound **9** (542 mg, 1.77 mmol) in anhydrous CH$_2$Cl$_2$ (5 mL) was added to the solvent and the mixture was further stirred for 15 min at room temperature. After completion of the reaction, the reaction mixture was diluted with CH$_2$Cl$_2$. The organic phase was washed with 10 (w/v)% sodium thiosulfate solution and brine, dried over Na$_2$SO$_4$, and concentrated *in vacuo*. The residue (1.28 g) was purified by column chromatography (*n*-hexane/AcOEt = 5:1) to give compound **4** (558 mg, 76%) as a white solid. IR $\nu_{max}$ (KBr): 3064, 3037, 3025, 2998, 2985, 2953, 2932, 2903, 2880, 1495, 1454, 1439, 1384, 1373, 1359, 1342, 1319, 1305 cm$^{-1}$. $^1$H-NMR (400 MHz, CDCl$_3$): $\delta$ = 7.38–7.30 (m, 5H), 6.11 (dtd, $J$ = 15.2, 8.1, 1.0 Hz, 1H), 5.72 (d, $J$ = 3.7 Hz, 1H), 5.65–5.59 (m, 1H), 4.75 (d, $J$ = 12.3 Hz, 1H), 4.62 (d, $J$ = 12.3 Hz, 1H), 4.57 (t, $J$ = 4.0 Hz, 1H), 4.49–4.45 (m, 1H), 3.87 (d, $J$ = 8.1 Hz, 2H), 3.48 (dd, $J$ = 9.0, 4.3 Hz, 1H), 1.61 (s, 3H), 1.36 (s, 3H). $^{13}$C-NMR (100 MHz, CDCl$_3$): $\delta$ = 137.4, 131.7, 130.3, 128.5, 128.1, 128.0, 113.0, 103.7, 81.5, 77.7, 77.5, 72.3, 26.7, 26.4, 4.2. HRMS (MALDI): calcd for C$_{17}$H$_{21}$O$_4$NaI [MNa$^+$] 439.0383, found 439.0377.

### 2′,3′-Anhydro-*N*³-[(*E*)-1-[(4 *R*)-3-*O*-benzyl-1,2-isopropylidene-α-ᴅ-erythrofuranos-4-yl]-1-propen-3-yl]-5′-*O*-tert-butyldiphenylsilyl-5-methyluridine (12)

Under N$_2$ atmosphere, NaH (12 mg, 0.31 mmol) was added quickly to a solution of compound **3**[75] (141 mg, 0.30 mmol) in anhydrous DMF (3 mL) at 0 °C. After the reaction mixture was stirred for 30 min at 0 °C, a solution of compound **4** (142 mg, 0.34 mmol) in anhydrous DMF (5 mL) was added dropwise to the solvent at 0 °C and the mixture was further stirred for 3 h at room temperature. After quenching with MeOH at room temperature, the whole mixture was extracted with AcOEt. The organic phase was washed with water and brine, dried over Na$_2$SO$_4$, and concentrated *in vacuo*. The residue (224 mg) was purified by column chromatography (*n*-hexane/AcOEt = 3:2) to give compound **12** (46 mg, 20%) as a yellow oil. IR $\nu_{max}$

(KBr): 2931, 2859, 1707, 1668, 1462, 1375, 1248 cm$^{-1}$. $^1$H-NMR (400 MHz, CDCl$_3$): $\delta$ = 7.64–7.61 (m, 4H), 7.46–7.30 (m, 11H), 7.24 (d, $J$ = 1.2 Hz, 1H), 5.96–5.89 (m, 1H), 5.71–5.77 (m, 2H), 5.67 (d, $J$ = 3.7 Hz, 1H), 4.70 (d, $J$ = 12.3 Hz, 1H), 4.61–4.55 (m, 2H), 4.52 (t, $J$ = 4.0 Hz, 1H), 4.47–4.42 (m, 2H), 4.32 (t, $J$ = 5.4 Hz, 1H), 4.07 (d, $J$ = 2.5 Hz, 1H), 3.91–3.81 (m, 3H), 3.48 (dd, $J$ = 9.0, 4.3 Hz, 1H), 1.72 (d, $J$ = 1.1 Hz, 3H), 1.58 (s, 3H), 1.34 (s, 3H), 1.06 (s, 9H). $^{13}$C-NMR (100 MHz, CDCl$_3$): $\delta$ = 162.8, 150.6, 137.6, 135.5, 135.4, 134.5, 132.8, 132.5, 131.3, 130.2, 130.1, 128.5, 127.8, 127.94, 127.92, 127.89 113.0, 110.1, 103.6, 87.4, 81.6, 81.3, 78.1, 77.8, 72.2, 63.4, 59.1, 58.9, 42.1, 26.9, 26.7, 26.5, 19.2, 13.1. HRMS (MALDI): calcd for C$_{43}$H$_{50}$N$_2$O$_9$NaSi [MNa$^+$] 789.3184, found 789.3178.

### 2,2′-Anhydro-3′-*O*-[(*E*)-1-[(4 *R*)-3-*O*-benzyl-1,2-isopropylidene-α-ᴅ-erythrofuranos-4-yl]-1-propen-3-yl]-5′-*O*-tert-butyldiphenylsilyl-5-methyluridine (10)

Under N$_2$ atmosphere, NaH (32 mg, 0.81 mmol) was added quickly to a solution of compound **3** (368 mg, 0.77 mmol) in anhydrous DMF (10 mL) at −15 °C using a low-temperature reactor. After the reaction mixture was stirred for 30 min at −15 °C, a solution of compound **4** (480 mg, 1.15 mmol) in anhydrous DMF (5 mL) was added dropwise to the solvent at −15 °C and the mixture was further stirred for 4 h at −15 °C. After quenching with MeOH at −15 °C, the whole mixture was extracted with AcOEt. The organic phase was washed with water and brine, dried over Na$_2$SO$_4$, and concentrated *in vacuo*. The residue (728 mg) was purified by column chromatography (AcOEt) to give compound **10** (407 mg, 69%) as a dark brown foam. IR $\nu_{max}$ (KBr): 2930, 2859, 1649, 1561, 1480, 1428, 1381, 1255 cm$^{-1}$. $^1$H-NMR (400 MHz, CDCl$_3$): $\delta$ = 7.58–7.54 (m, 4H), 7.45–7.27 (m, 11H), 7.12 (d, $J$ = 1.3 Hz, 1H), 6.04 (d, $J$ = 5.8 Hz, 1H), 5.93 (dtd, $J$ = 16.0, 5.3, 0.8 Hz, 1H), 5.74–5.68 (m, 2H), 5.18 (dd, $J$ = 5.9, 0.9 Hz, 1H), 4.75 (d, $J$ = 12.1 Hz, 1H), 4.59–4.56 (m, 2H), 4.50 (dd, $J$ = 8.7, 6.4 Hz, 1H), 4.32–4.31 (m, 1H), 4.26 (ddd, $J$ = 7.0, 5.7, 2.7 Hz, 1H), 4.07 (d, $J$ = 5.3 Hz, 2H), 3.57–3.49 (m, 2H), 3.43 (dd, $J$ = 11.1, 7.1 Hz, 1H), 1.93 (s, 3H), 1.61 (s, 3H), 1.36 (s, 3H), 1.01 (s, 9H). $^{13}$C-NMR (100 MHz, CDCl$_3$): $\delta$ = 172.0, 159.0, 137.4, 135.44, 135.43, 132.7, 132.3, 130.4, 130.10, 130.07, 129.8, 128.8, 128.4, 128.1, 128.01, 127.97, 127.91, 119.6, 113.0, 103.8, 90.0, 86.4, 85.9, 82.7, 81.9, 77.8, 77.4, 72.3, 70.3, 62.6, 26.8, 26.7, 26.5, 19.2, 14.1. HRMS (MALDI): calcd for C$_{43}$H$_{50}$N$_2$O$_9$NaSi [MNa$^+$] 789.3184, found 789.3178.

### 1-[3-*O*-[(*E*)-1-[(4 *R*)-3-*O*-Benzyl-1,2-isopropylidene-α-ᴅ-erythrofuranos-4-yl]-1-propen-3-yl]-5-*O*-tert-butyldiphenylsilyl-β-ᴅ-arabinofuranosyl]thymine (1) and 1-[3-*O*-[(*E*)-1-[(4 *R*)-3-*O*-Benzyl-1,2-isopropylidene-α-ᴅ-erythrofuranos-4-yl]-1-propen-3-yl]-β-ᴅ-arabinofuranosyl]thymine (11)

1.0 M NaOH aq. (0.28 mL, 0.28 mmol) was added to a solution of compound **10** (423 mg, 0.55 mmol) in EtOH (10 mL) at room temperature. The reaction mixture was refluxed for 20 h. After being neutralized with sat. NH$_4$Cl aq. at room temperature, the whole mixture was extracted with AcOEt. The organic phase was washed with water and brine, dried over Na$_2$SO$_4$, and concentrated *in vacuo*. The residue (344 mg) was purified by column chromatography (*n*-hexane/AcOEt = 1:2) to give compound **1** (253 mg, 58%) as a white foam and compound **11** (43 mg, 14%) as a white foam. Compound **1**: IR $\nu_{max}$ (KBr): 3384, 3184, 3029, 2932, 2859, 1699, 1666, 1473, 1428, 1384, 1282 cm$^{-1}$. $^1$H-NMR (500 MHz, CDCl$_3$): $\delta$ = 8.03 (s, 1H), 7.70–7.65 (m, 4H), 7.56 (d, $J$ = 1.2 Hz, 1H), 7.51–7.41 (m, 6H), 7.34–7.28 (m, 5H), 6.08 (d, $J$ = 3.0 Hz, 1H), 5.89–5.95 (m, 1H), 5.73–5.67 (m, 2H), 4.73 (d, $J$ = 12.1 Hz, 1H), 4.58–4.55 (m, 2H), 4.50–4.47 (m, 1H), 4.27 (ddd, $J$ = 10.0, 3.1, 1.2 Hz, 1H), 4.17–4.13 (m, 1H), 4.10 (q, $J$ = 2.1 Hz, 1H), 4.06–3.98 (m, 4H), 3.76 (dd, $J$ = 11.6, 2.0 Hz, 1H), 3.50 (dd, $J$ = 9.0, 4.3 Hz, 1H), 1.75 (d, $J$ = 1.0 Hz, 3H), 1.61 (s, 3H), 1.36 (s, 3H), 1.09 (s, 9H). $^{13}$C-NMR (126 MHz, CDCl$_3$): $\delta$ = 163.9, 150.2, 137.6, 137.5, 135.6, 135.4, 131.9, 131.8, 130.4, 130.3, 129.8, 129.6, 128.5, 128.1, 128.0, 113.0, 108.9, 103.8, 86.5, 84.6, 83.2, 82.0, 78.0, 77.6, 72.9, 72.3, 69.9, 64.1, 26.8, 26.5, 19.1, 12.2. HRMS (MALDI): calcd for C$_{43}$H$_{52}$N$_2$O$_{10}$NaSi [MNa$^+$] 807.3289, found 807.3283. Compound **11**: IR $\nu_{max}$ (KBr): 3382, 3032, 2987, 2928, 1697, 1469, 1382, 1281 cm$^{-1}$. $^1$H-NMR (500 MHz, CDCl$_3$): $\delta$ = 8.86 (s, 1H),

7.56 (d, $J$ = 1.1 Hz, 1H), 7.36–7.29 (m, 5H), 6.05 (d, $J$ = 3.5 Hz, 1H), 5.93 (dt, $J$ = 16.3, 5.4 Hz, 1H), 5.69–5.74 (m, 2H), 4.75 (d, $J$ = 12.0 Hz, 1H), 4.60–4.56 (m, 2H), 4.52–4.49 (m, 1H), 4.42 (d, $J$ = 7.5 Hz, 1H), 4.33 (d, $J$ = 7.9 Hz, 1H), 4.20 (dd, $J$ = 13.7, 5.6 Hz, 1H), 4.12–4.07 (m, 2H), 4.00–3.96 (m, 2H), 3.82 (dt, $J$ = 11.7, 2.8 Hz, 1H), 3.53 (dd, $J$ = 9.0, 4.3 Hz, 1H), 2.56 (t, $J$ = 4.1 Hz, 1H), 1.85 (s, 3H), 1.61 (s, 3H), 1.36 (s, 3H). $^{13}$C-NMR (126 MHz, CDCl$_3$): $\delta$ = 165.4, 150.5, 138.9, 137.5, 130.1, 130.0, 128.5, 128.1, 128.0, 113.0, 108.2, 103.8, 87.4, 84.0, 82.0, 78.1, 77.5, 72.5, 72.3, 69.5, 62.6, 26.7, 26.5, 12.3. HRMS (MALDI): calcd for C$_{27}$H$_{34}$N$_2$O$_{10}$Na [MNa$^+$] 569.2111, found 569.2106.

### (4*R*)-3-*O*-Benzyl-4-ethynyl-1,2-isopropylidene-α-D-erythrofuranose (13)

NaIO$_4$ (620 mg, 2.9 mmol) was added to a solution of compound **7** (692 mg, 2.23 mmol) in H$_2$O (10 mL) and THF (10 mL) at room temperature. The reaction mixture was stirred for 1 h at room temperature. The reaction mixture was filtered by vacuum filtration, and the filtrate was extracted with AcOEt. The organic phase was washed with water and brine, dried over Na$_2$SO$_4$, and concentrated *in vacuo*. The obtained residue (591 mg) was dissolved in anhydrous MeOH (20 mL) under N$_2$ atmosphere at room temperature. Dimethyl (1-diazo-2-oxopropyl)phosphonate (Ohira-Bestmann reagent) (0.37 mL, 2.5 mmol) and K$_2$CO$_3$ (616 mg, 4.5 mmol) was added to the solution at room temperature. The reaction mixture was stirred for 14 h at room temperature. After addition of sat. NH$_4$Cl aq. at room temperature, the whole mixture was extracted with AcOEt. The organic phase was washed with water and brine, dried over Na$_2$SO$_4$, and concentrated *in vacuo*. The residue (583 mg) was purified by column chromatography (*n*-hexane/AcOEt = 9:1) to give compound **13** (500 mg, 82%, two steps from compound **7**) as a white solid. IR ν$_{max}$ (KBr): 3276, 2989, 2936, 2890, 2130, 1455, 1375, 1337, 1314, 1291, 1247 cm$^{-1}$. $^1$H-NMR (400 MHz, CDCl$_3$): $\delta$ = 7.43–7.30 (m, 5H), 5.75 (d, $J$ = 3.6 Hz, 1H), 4.85 (d, $J$ = 12.3 Hz, 1H), 4.78 (d, $J$ = 12.3 Hz, 1H), 4.67 (dd, $J$ = 9.0, 2.1 Hz, 1H), 4.53 (t, $J$ = 4.0 Hz, 1H), 3.88 (dd, $J$ = 9.0, 4.3 Hz, 1H), 2.56 (d, $J$ = 2.1 Hz, 1H), 1.59 (s, 3H), 1.34 (s, 3H). $^{13}$C-NMR (100 MHz, CDCl$_3$): $\delta$ = 137.2, 128.5, 128.2, 113.3, 103.7, 81.5, 80.3, 77.7, 75.1, 72.5, 68.0, 26.7, 26.3. HRMS (MALDI): calcd for C$_{16}$H$_{18}$O$_4$Na [MNa$^+$] 297.1103, found 297.1097.

### 3-[(4*R*)-3-*O*-Benzyl-1,2-isopropylidene-α-D-erythrofuranos-4-yl]-2-propyn-1-ol (14)

Under N$_2$ atmosphere, *n*-BuLi (1.58 M in THF, 13 mL, 20 mmol) was added dropwise to a solution of compound **13** (5.0 g, 18 mmol) in anhydrous THF (100 mL) at −78 °C. After the reaction mixture was stirred for 1 h at −78 °C, gaseous formaldehyde generated from (HCHO)$_n$ (5.4 g, 0.18 mol) heated by heat gun was passed over the solution in a slow stream of N$_2$ and the mixture was further stirred for 1 h at room temperature. After quenching with sat. NH$_4$Cl aq. at room temperature, the whole mixture was extracted with AcOEt. The organic phase was washed with water and brine, dried over Na$_2$SO$_4$, and concentrated *in vacuo*. The residue (5.7 g) was purified by column chromatography (*n*-hexane/AcOEt = 2:1) to give compound **14** (4.7 g, 85%) as a colorless oil. IR ν$_{max}$ (KBr): 3471, 2986, 2943, 2873, 1457, 1375 cm$^{-1}$. $^1$H-NMR (400 MHz, CDCl$_3$): $\delta$ = 7.44–7.30 (m, 5H), 5.74 (d, $J$ = 3.7 Hz, 1H), 4.86 (d, $J$ = 12.3 Hz, 1H), 4.76–4.69 (m, 2H), 4.55 (t, $J$ = 4.0 Hz, 1H), 4.31 (dd, $J$ = 6.3, 1.5 Hz, 2H), 3.85 (dd, $J$ = 9.0, 4.3 Hz, 1H), 1.59 (s, 3H), 1.52 (t, $J$ = 6.3 Hz, 1H), 1.35 (s, 3H). $^{13}$C-NMR (126 MHz, CDCl$_3$): $\delta$ = 137.3, 128.5, 128.2, 128.1, 113.3, 103.7, 85.3, 82.1, 81.6, 77.5, 72.5, 68.2, 51.1, 26.7, 26.3. HRMS (MALDI): calcd for C$_{17}$H$_{20}$O$_5$Na [MNa$^+$] 327.1209, found 327.1203.

### (4*R*)-3-*O*-Benzyl-4-(3-iodoprop-1-yn-1-yl)-1,2-isopropylidene-α-D-erythrofuranose (5)

Under N$_2$ atmosphere, I$_2$ (5.5 g, 22 mmol) was added to a solution of Ph$_3$P (5.6 g, 22 mmol) and imidazole (1.5 g, 22 mmol) in anhydrous CH$_2$Cl$_2$ (70 mL) at room temperature. After the reaction mixture was stirred for 15 min at room temperature, a solution of compound **14** (4.7 g, 15 mmol) in anhydrous CH$_2$Cl$_2$ (30 mL) was added to the solvent and the mixture was further stirred for 15 min at room temperature. After completion of the

reaction, the reaction mixture was diluted with CH$_2$Cl$_2$. The organic phase was washed with 10 (w/v)% sodium thiosulfate solution and brine, dried over Na$_2$SO$_4$, and concentrated *in vacuo*. The residue (13.0 g) was purified by column chromatography (*n*-hexane/AcOEt = 5:1) to give compound **5** (5.8 g, 91%) as a yellow oil.

IR ν$_{max}$ (KBr): 3061, 3030, 2985, 2932, 2896, 1496, 1454, 1373, 1338, 1313, 1290 cm$^{-1}$. $^1$H-NMR (300 MHz, CDCl$_3$): $\delta$ = 7.46–7.30 (m, 5H), 5.72 (d, $J$ = 3.7 Hz, 1H), 4.85 (d, $J$ = 12.5 Hz, 1H), 4.78 (d, $J$ = 12.3 Hz, 1H), 4.69 (dt, $J$ = 8.9, 2.1 Hz, 1H), 4.53 (t, $J$ = 3.9 Hz, 1H), 3.83 (dd, $J$ = 8.9, 4.3 Hz, 1H), 3.73 (d, $J$ = 2.0 Hz, 2H), 1.59 (s, 3H), 1.34 (s, 3H). $^{13}$C-NMR (126 MHz, CDCl$_3$): $\delta$ = 137.2, 128.5, 128.2, 128.1, 113.3, 103.6, 83.7, 81.4, 81.3, 77.7, 72.5, 68.3, 26.7, 26.3, −19.4 (propargylic carbon). HRMS (MALDI): calcd for C$_{17}$H$_{19}$O$_4$NaI [MNa$^+$] 437.0226, found 437.0220.

### 2,2′-Anhydro-3′-*O*-[1-[(4*R*)-3-*O*-benzyl-1,2-isopropylidene-α-D-erythrofuranos-4-yl]-1-propyn-3-yl]-5′-*O*-*tert*-butyldiphenylsilyl-5-methyluridine (15)

Under N$_2$ atmosphere, NaH (308 mg, 7.7 mmol) was added quickly to a solution of compound **3** (3.5 g, 7.3 mmol) in anhydrous DMF (60 mL) at −15 °C using a low-temperature reactor. After the reaction mixture was stirred for 30 min at −15 °C, a solution of compound **5** (4.6 g, 11 mmol) in anhydrous DMF (20 mL) was added dropwise to the solvent at −15 °C and the mixture was further stirred for 4 h at −15 °C. After quenching with MeOH at −15 °C, the whole mixture was extracted with AcOEt. The organic phase was washed with water and brine, dried over Na$_2$SO$_4$, and concentrated *in vacuo*. The residue (6.9 g) was purified by column chromatography (AcOEt) to give compound **16** (4.3 g, 77%) as a dark brown foam. IR ν$_{max}$ (KBr): 2961, 2936, 2854, 1644, 1561, 1480 cm$^{-1}$. $^1$H-NMR (500 MHz, CDCl$_3$): $\delta$ = 7.58–7.54 (m, 4H), 7.28–7.45 (m, 11H), 7.10 (d, $J$ = 1.3 Hz, 1H), 6.01 (d, $J$ = 5.7 Hz, 1H), 5.71 (d, $J$ = 3.6 Hz, 1H), 5.33 (d, $J$ = 6.2 Hz, 1H), 4.82 (d, $J$ = 12.0 Hz, 1H), 4.73–4.68 (m, 2H), 4.55 (t, $J$ = 3.9 Hz, 1H), 4.40 (d, $J$ = 2.3 Hz, 1H), 4.34 (dd, $J$ = 15.8, 1.6 Hz, 1H), 4.29–4.22 (m, 2H), 3.88 (dd, $J$ = 9.0, 4.2 Hz, 1H), 3.55 (dd, $J$ = 11.1, 5.9 Hz, 1H), 3.43 (dd, $J$ = 11.1, 7.2 Hz, 1H), 1.93 (d, $J$ = 1.2 Hz, 3H), 1.34 (s, 3H), 1.01 (s, 9H). $^{13}$C-NMR (100 MHz, CDCl$_3$): $\delta$ = 172.0, 159.0, 137.2, 135.44, 135.42, 132.7, 132.3, 130.12, 130.08, 129.8, 128.5, 128.11, 128.09, 128.0, 127.9, 119.6, 113.4, 103.9, 90.0, 86.2, 85.9, 84.5, 82.7, 81.9, 81.1, 77.4, 72.6, 68.2, 62.6, 58.4, 26.8, 26.7, 26.4, 19.1, 14.1. HRMS (MALDI): calcd for C$_{43}$H$_{48}$N$_2$O$_9$NaSi [MNa$^+$] 787.3027, found 787.3021.

### 1-[3-*O*-[1-[(4*R*)-3-*O*-benzyl-1,2-isopropylidene-α-D-erythrofuranos-4-yl]-1-propyn-3-yl]-5-*O*-*tert*-butyldiphenylsilyl-β-D-arabinofuranosyl]thymine (2) and 1-[3-*O*-[1-[(4*R*)-3-*O*-benzyl-1,2-isopropylidene-α-D-erythrofranos-4-yl]-1-propyn-3-yl]-β-D-arabinofuranosyl]thymine (16)

1.1 M NaOH aq. (2.6 mL, 2.8 mmol) was added to a solution of compound **15** (4.3 g, 5.7 mmol) in EtOH (80 mL) at room temperature. The reaction mixture was refluxed for 21 h. After being neutralized with sat. NH$_4$Cl aq. at room temperature, the whole mixture was extracted with AcOEt. The organic phase was washed with sat. NH$_4$Cl aq. and brine, dried over Na$_2$SO$_4$, and concentrated *in vacuo*. The residue (4.4 g) was purified by column chromatography (*n*-hexane/AcOEt = 1:2) to give compound **2** (3.2 g, 72%) as a yellow foam and compound **16** (632 mg, 20%) as a white foam, respectively. Compound **2**: IR ν$_{max}$ (KBr): 3351, 3173, 3031, 2934, 2895, 2859, 1698, 1666, 1473, 1428, 1385, 1373, 1311, 1283 cm$^{-1}$. $^1$H-NMR (500 MHz, CDCl$_3$): $\delta$ = 8.32 (brs, 1H), 7.70–7.66 (m, 4H), 7.53–7.28 (m, 12H), 6.08 (d, $J$ = 3.2 Hz, 1H), 5.71 (d, $J$ = 3.6 Hz, 1H), 4.81 (d, $J$ = 12.2 Hz, 1H), 4.71–4.68 (m, 2H), 4.53 (t, $J$ = 3.9 Hz, 1H), 4.37–4.27 (m, 3H), 4.20 (s, 1H), 4.15 (d, $J$ = 9.4 Hz, 1H), 4.09 (q, $J$ = 2.4 Hz, 1H), 4.00 (dd, $J$ = 11.6, 2.8 Hz, 1H), 3.84–3.78 (m, 2H), 1.72 (s, 3H), 1.58 (s, 3H), 1.35 (s, 3H), 1.09 (s, 9H). $^{13}$C-NMR (126 MHz, CDCl$_3$): $\delta$ = 164.7, 150.3, 138.4, 137.2, 135.5, 135.4, 132.4, 132.2, 130.2, 130.1, 128.5, 128.11, 128.08, 128.02, 128.00, 113.3, 108.4, 103.7, 86.5, 83.9, 83.5, 83.0, 82.3, 81.6, 77.6, 72.8, 72.5, 68.2, 63.6, 57.5, 26.8, 26.4, 19.2, 12.1. HRMS (MALDI): calcd for C$_{43}$H$_{50}$N$_2$O$_{10}$NaSi [MNa$^+$] 805.3133, found 805.3127. Compound **17**: IR ν$_{max}$ (KBr): 3340, 3031, 2987,

2932, 1698, 1668, 1476, 1456, 1375, 1282 cm$^{-1}$. $^{1}$H-NMR (300 MHz, CDCl$_3$): $\delta$ = 9.96 (s, 1H), 7.65 (d, $J$ = 1.1 Hz, 1H), 7.43–7.28 (m, 5H), 6.06 (d, $J$ = 3.5 Hz, 1H), 5.75 (d, $J$ = 3.6 Hz, 1H), 5.05 (d, $J$ = 5.6 Hz, 1H), 4.83 (d, $J$ = 12.2 Hz, 1H), 4.76–4.70 (m, 2H), 4.67–4.63 (m, 1H), 4.54 (t, $J$ = 4.0 Hz, 1H), 4.43 (dd, $J$ = 16.1, 1.5 Hz, 1H), 4.35 (dd, $J$ = 16.1, 1.5 Hz, 1H), 4.21–4.19 (m, 1H), 4.14 (q, $J$ = 2.9 Hz, 1H), 3.99–3.81 (m, 3H), 2.88 (t, $J$ = 5.0 Hz, 1H), 1.78 (d, $J$ = 1.0 Hz, 3H), 1.59 (s, 3H), 1.34 (s, 3H). $^{13}$C-NMR (126 MHz, CDCl$_3$): $\delta$ = 165.6, 150.5, 139.0, 137.2, 128.5, 128.14, 128.12, 113.3, 108.1, 103.8, 87.2, 83.8, 83.7, 83.6, 82.4, 81.8, 77.5, 72.6, 72.3, 68.2, 62.4, 57.5, 26.7, 26.3, 12.3. HRMS (MALDI): calcd for C$_{27}$H$_{32}$N$_2$O$_{10}$Na [MNa$^+$] 567.1955, found 567.1949.

### (4a*S*,5 *R*,7 *R*,7a*R*)-(*Z*)-3-[(4 *R*)-3-*O*-Benzyl-1,2-isopropylidene-α-ᴅ-erythrofuranos-4-yl]methylene-7-*tert*-butyldiphenylsilyloxymethyl-5-(thymin-1-yl)-hexahydrofuro[3,4-*b*][1,4]dioxine (17) and (5a*S*,6 *R*,8 *R*,8a*R*)-(*E*)-4-[(4 *R*)-3-*O*-Benzyl-1,2-isopropylidene-α-ᴅ-erythrofuranos-4-yl]-8-*tert*-butyldiphenylsilyloxymethyl-6-(thymin-1-yl)-5a,6,8,8a-tetrahydro-2*H*-furo[3,4-*b*][1,4]dioxepine (18)

Under N$_2$ atmosphere, [(Ph$_3$PAu)$_3$O]BF$_4$ (287 mg, 0.19 mmol) was added to a solution of compound **2** (3.0 g, 3.9 mmol) in anhydrous 1,2-dichloroethane (40 mL) at room temperature. The reaction mixture was refluxed for 23 h. After completion of the reaction, the resulting mixture was filtrated by short column chromatography (*n*-hexane/AcOEt = 1:2) and the filtrate was concentrated *in vacuo*. The residue (2.8 g) was purified by column chromatography (*n*-hexane/AcOEt = 1:1) to give a mixture of compound **17** and compound **18** (2.5 g, **17**:**18** = 1:1) as a white foam. This mixture was purified by column chromatography (CHCl$_3$/CH$_2$Cl$_2$/CH$_3$CN/*n*-hexane = 1:1:1:1) two times to give compound **17** (1.1 g, 38%) as a white foam and compound **18** (1.1 g, 38%) as a white foam, respectively. Compound **17**: IR ν$_{max}$ (KBr): 3068, 2929, 2901, 2860, 1689, 1470, 1428, 1372, 1267 cm$^{-1}$. $^{1}$H-NMR (500 MHz, CDCl$_3$): $\delta$ = 7.94 (s, 1H), 7.65–7.68 (m, 4H), 7.44–7.29 (m, 12H), 6.24 (d, $J$ = 6.8 Hz, 1H), 5.69 (d, $J$ = 3.7 Hz, 1H), 4.89 (t, $J$ = 9.0 Hz, 1H), 4.76 (d, $J$ = 12.3 Hz, 1H), 4.68 (d, $J$ = 8.9 Hz, 1H), 4.58–4.55 (m, 2H), 4.36–4.33 (m, 2H), 4.22–4.15 (m, 2H), 4.13–4.09 (m, 1H), 3.93–3.88 (m, 2H), 3.51 (dd, $J$ = 9.1, 4.2 Hz, 1H), 1.61 (s, 3H), 1.53 (d, $J$ = 1.0 Hz, 3H), 1.35 (s, 3H), 1.10 (s, 9H). $^{13}$C-NMR (126 MHz, CDCl$_3$): $\delta$ = 163.3, 152.2, 150.0, 137.8, 135.5, 135.3, 135.0, 133.0, 132.5, 130.11, 130.09, 128.4, 128.0, 127.91, 127.90, 113.2, 110.6, 107.6, 103.7, 81.9, 79.8, 79.7, 77.8, 77.5, 73.4, 72.1, 71.0, 67.3, 61.7, 27.0, 26.7, 26.6, 19.5, 12.2, 1.9. HRMS (MALDI): calcd for C$_{43}$H$_{50}$N$_2$O$_{10}$NaSi [MNa$^+$] 805.3133, found 805.3127. Compound **18**: IR ν$_{max}$ (KBr): 3191, 3068, 2933, 2859, 1686, 1471, 1428, 1372, 1330, 1304, 1269 cm$^{-1}$. $^{1}$H-NMR (500 MHz, CD$_3$CN): $\delta$ = 8.98 (brs, 1H), 7.72–7.65 (m, 4H), 7.47–7.24 (m, 12H), 6.42 (d, $J$ = 7.4 Hz, 1H), 5.64 (d, $J$ = 3.5 Hz, 1H), 5.14 (dd, $J$ = 6.0, 2.4 Hz, 1H), 4.59 (t, $J$ = 3.9 Hz, 1H), 4.46–4.42 (m, 2H), 4.39–4.33 (m, 2H), 4.23–4.17 (m, 2H), 4.13 (t, $J$ = 8.5 Hz, 1H), 4.08 (dd, $J$ = 11.8, 1.5 Hz, 1H), 3.98–3.91 (m, 2H), 3.74 (dd, $J$ = 8.6, 4.3 Hz, 1H), 1.47 (s, 3H), 1.44 (s, 3H), 1.29 (s, 3H), 1.05 (s, 9H). $^{13}$C-NMR (126 MHz, CDCl$_3$): $\delta$ = 163.3, 152.1, 150.0, 137.5, 135.5, 135.3, 133.0, 132.5, 130.12, 130.07, 128.4, 128.0, 127.9, 127.6, 113.3, 110.2, 108.8, 104.0, 84.7, 81.0, 80.0, 79.6, 78.5, 78.0, 77.2, 72.9, 66.8, 61.4, 27.04, 27.02, 26.4, 19.5, 12.1. HRMS (MALDI): calcd for C$_{43}$H$_{50}$N$_2$O$_{10}$NaSi [MNa$^+$] 805.3133, found 805.3127.

### (4a*S*,5 *R*,7 *R*,7a*R*)-3-[(4 *R*)-3-*O*-Benzyl-1,2-isopropylidene-α-ᴅ-erythrofuranos-4-yl]methyl-7-*tert*-butyldiphenylsilyloxymethyl-5-(thymin-1-yl)-4a,5,7,7a-tetrahydrofuro[3,4-*b*][1,4]dioxine (19)

10 (w/w) % Pd/C (en) (66 mg) was added to a solution of compound **17** (51 mg, 65 μmol) in THF (1 mL) at room temperature. After the reaction mixture was stirred for 24 h at room temperature under H$_2$ atmosphere at 1 atm, the mixture was filtered by Celite$^{®}$, and the filtrate was concentrated *in vacuo*. The residue (50 mg) was purified by column chromatography (*n*-hexane/AcOEt = 3:2) to give compound **19** (25 mg, 49%) as a white foam. IR ν$_{max}$ (KBr): 3525, 3188, 3070, 2931, 2859, 1686, 1470, 1428, 1372, 1326, 1307, 1267 cm$^{-1}$. $^{1}$H-NMR (400 MHz, CDCl$_3$): $\delta$ = 8.02 (s, 1H), 7.68–7.64 (m, 4H), 7.46–7.30 (m, 12H), 6.34 (d, $J$ = 6.8 Hz, 1H), 5.99 (s, 1H), 5.70 (d,

$J$ = 3.8 Hz, 1H), 4.76 (d, $J$ = 11.6 Hz, 1H), 4.56–4.51 (m, 2H), 4.25 (dd, $J$ = 8.5, 6.8 Hz, 1H), 4.18–4.08 (m, 3H), 3.98–3.93 (m, 2H), 3.48 (dd, $J$ = 9.0, 4.3 Hz, 1H), 2.42 (dd, $J$ = 15.5, 4.3 Hz, 1H), 2.29 (dd, $J$ = 15.3, 6.9 Hz, 1H), 1.55 (s, 3H), 1.52 (d, $J$ = 1.1 Hz, 3H), 1.35 (s, 3H), 1.10 (s, 9H). $^{13}$C-NMR (126 MHz, CDCl$_3$): $\delta$ = 163.2, 150.1, 137.6, 135.53, 135.48, 135.3, 135.0, 133.0, 132.4, 130.1, 130.0, 128.5, 128.11, 128.07, 128.0, 124.1, 112.9, 110.6, 104.0, 81.4, 79.8, 77.8, 77.23, 77.16, 74.9, 72.3, 71.2, 61.8, 32.0, 27.0, 26.7, 26.6, 19.5, 12.2. HRMS (MALDI): calcd for C$_{43}$H$_{50}$N$_2$O$_{10}$NaSi [MNa$^+$] 805.3133, found 805.3127.

### (4a*S*,5 *R*,7 *R*,7a*R*)-3-[(4 *R*)-1,2-Isopropylidene-α-ᴅ-erythrofuranos-4-yl]methyl-7-*tert*-butyldiphenylsilyloxymethyl-5-(thymin-1-yl)-4a,5,7,7a-tetrahydrofuro[3,4-*b*][1,4]dioxine (20)

20 (w/w) % Pd(OH)$_2$/C (65 mg) was added to a solution of compound **17** (326 mg, 0.42 mmol) in THF (8 mL) at room temperature. After the reaction mixture was stirred for 25 h at room temperature under H$_2$ atmosphere at 1 atm, the mixture was filtered, and the filtrate was concentrated *in vacuo*. The residue (309 mg) was purified by column chromatography (*n*-hexane/AcOEt = 1:3) to give compound **20** (244 mg, 85%) as a white foam. IR ν$_{max}$ (KBr): 3450, 3322, 3198, 3071, 2932, 2860, 1687, 1472, 1428, 1373, 1326, 1269 cm$^{-1}$. $^{1}$H-NMR (500 MHz, CDCl$_3$): $\delta$ = 8.35 (s, 1H), 7.69–7.65 (m, 4H), 7.46–7.38 (m, 7H), 6.37 (d, $J$ = 6.7 Hz, 1H), 6.03 (s, 1H), 5.74 (d, $J$ = 3.9 Hz, 1H), 4.53 (t, $J$ = 4.5 Hz, 1H), 1.10 (s, 9H), 4.36 (dd, $J$ = 8.5, 6.7 Hz, 1H), 4.18–4.12 (m, 2H), 4.03 (dt, $J$ = 9.8, 2.5 Hz, 1H), 3.98 (dd, $J$ = 12.2, 3.2 Hz, 1H), 3.85 (dt, $J$ = 8.8, 6.0 Hz, 1H), 3.64 (ddd, $J$ = 10.0, 9.0, 5.1 Hz, 1H), 2.65 (d, $J$ = 10.2 Hz, 1H), 2.45 (dd, $J$ = 15.2, 6.6 Hz, 1H), 2.35 (dd, $J$ = 15.1, 5.4 Hz, 1H), 1.54–1.51 (m, 6H), 1.35 (s, 3H). $^{13}$C-NMR (126 MHz, CDCl$_3$): $\delta$ = 163.4, 150.5, 135.5, 135.4, 135.2, 135.0, 133.0, 132.5, 130.10, 130.06, 127.97, 127.96, 124.1, 112.6, 110.7, 103.6, 80.1, 78.7, 78.0, 77.2, 77.0, 75.1, 71.2, 61.8, 31.9, 27.0, 26.5, 19.5, 12.1. HRMS (MALDI): calcd for C$_{36}$H$_{44}$N$_2$O$_{10}$NaSi [MNa$^+$] 715.2663, found 715.2657.

### (4a*S*,5 *R*,7 *R*,7a*R*)-3-[(2 *R*,3 *R*,4 *R*,5 *R*)-3,4-Diacethyloxy-2-(thymin-1-yl)-tetrahydrofuran-5-yl]methyl-7-*tert*-butyldiphenylsilyloxymethyl-5-(thymin-1-yl)-4a,5,7,7a-tetrahydrofuro[3,4-*b*][1,4]dioxine (21)

Compound **20** (137 mg, 0.20 mmol) was dissolved in 60 (v/v)% AcOH aq. (4 mL) at room temperature, and the solution was refluxed for 2 h. After being neutralized with sat. NaHCO$_3$ aq. at room temperature, the whole mixture was extracted with AcOEt. The organic phase was washed with sat. NaHCO$_3$ aq. and brine, dried over Na$_2$SO$_4$, and concentrated *in vacuo*. The obtained residue (86 mg) was dissolved in CH$_2$Cl$_2$ (4 mL), and Ac$_2$O (0.28 mL, 3.0 mmol), Et$_3$N (0.27 mL, 2.0 mmol) and DMAP (12 mg, 99 μmol) were added at room temperature. After the reaction mixture was stirred for 1 h at room temperature, the whole mixture was extracted with AcOEt. The organic phase was washed with sat. NaHCO$_3$ aq., water and brine, dried over Na$_2$SO$_4$, and concentrated *in vacuo*. The obtained residue (101 mg) was dissolved in anhydrous 1,2-dichloroethane (2 mL), and thymine (33 mg, 0.26 mmol) and BSA (0.16 ml, 0.65 mmol) were added under N$_2$ atmosphere at room temperature. The solution was refluxed for 25 min and then cooled to 0 °C. TMSOTf (47 μL, 0.26 mmol) was added, and the reaction mixture was refluxed for 20 h. After quenching with sat. NaHCO$_3$ aq. at room temperature, the whole mixture was extracted with AcOEt. The organic phase was washed with water and brine, dried over Na$_2$SO$_4$, and concentrated *in vacuo*. The residue (109 mg) was purified by column chromatography (CHCl$_3$/MeOH = 19:1) to give compound **21** (65 mg, 39%, 3 steps from compound **20**) as a white foam. IR ν$_{max}$ (KBr): 3185, 3069, 2930, 2858, 1748, 1696, 1471, 1428, 1372, 1269 cm$^{-1}$. $^{1}$H-NMR (500 MHz, CDCl$_3$): $\delta$ = 8.51 (s, 1H), 8.47 (s, 1H), 7.69–7.65 (m, 4H), 7.36–7.49 (m, 7H), 7.15 (d, $J$ = 1.1 Hz, 1H), 6.36 (d, $J$ = 6.5 Hz, 1H), 6.04 (s, 1H), 5.97 (d, $J$ = 6.0 Hz, 1H), 5.33 (t, $J$ = 6.0 Hz, 1H), 5.27 (dd, $J$ = 5.9, 4.5 Hz, 1H), 4.32 (dd, $J$ = 8.4, 6.6 Hz, 1H), 4.20–4.11 (m, 3H), 4.05 (dt, $J$ = 9.7, 2.5 Hz, 1H), 3.97 (dd, $J$ = 12.3, 3.2 Hz, 1H), 2.62 (dd, $J$ = 15.5, 4.6 Hz, 1H), 2.42 (dd, $J$ = 15.2, 7.5 Hz, 1H), 2.11 (s, 3H), 2.09 (s, 3H), 1.95 (d, $J$ = 0.9 Hz, 3H), 1.54

(d, $J = 0.8$ Hz, 3H), 1.10 (s, 9H). $^{13}$C-NMR (126 MHz, CDCl$_3$): $\delta = 170.1$, 169.6, 163.7, 163.5, 150.4, 150.3, 135.7, 135.6, 135.3, 134.6, 134.3, 132.9, 132.3, 130.14, 130.09, 128.0, 124.8, 111.9, 110.6, 87.4, 80.2, 79.4, 78.0, 77.7, 72.7, 72.3, 72.0, 71.0, 61.8, 32.9, 27.0, 20.6, 20.5, 19.5, 12.6, 12.2. HRMS (MALDI): calcd for C$_{42}$H$_{48}$N$_4$O$_{13}$NaSi [MNa$^+$] 867.2885, found 867.2879.

### (4a*S*,5*R*,7*R*,7a*R*)-3-[(2*R*,3*R*,4*R*,5*R*)-3,4-Dihydroxy-2-(thymin-1-yl)-tetrahydrofuran-5-yl]methyl-7-*tert*-butyldiphenylsilyloxymethyl-5-(thymin-1-yl)-4a,5,7,7a-tetrahydrofuro[3,4-*b*][1,4]dioxine (22)

28% NaOMe in MeOH (9.4 µL, 49 µmol) was added to a solution of compound **21** (81 mg, 96 µmol) in MeOH (1.5 mL) at 0 °C. The reaction mixture was stirred for 22 h at room temperature. After quenching with sat. NH$_4$Cl aq. at room temperature, the whole mixture was extracted with AcOEt. The organic phase was washed with sat. NH$_4$Cl aq. and brine, dried over Na$_2$SO$_4$, and concentrated *in vacuo*. The residue (87 mg) was purified by column chromatography (CHCl$_3$/MeOH = 19:1) to give compound **22** (59 mg, 81%) as a white foam. IR $\nu_{max}$ (KBr): 3396, 3203, 3067, 2928, 2859, 1687, 1472, 1428, 1269 cm$^{-1}$. $^1$H-NMR (400 MHz, CDCl$_3$): $\delta = 10.19$ (s, 1H), 9.82 (s, 1H), 7.69–7.65 (m, 4H), 7.50 (d, $J = 1.0$ Hz, 1H), 7.47–7.36 (m, 6H), 1.10 (s, 9H), 7.28 (d, $J = 0.9$ Hz, 1H), 6.32 (d, $J = 6.4$ Hz, 1H), 6.04 (s, 1H), 5.70 (d, $J = 4.4$ Hz, 1H), 4.90 (s, 1H), 4.39 (dd, $J = 7.6$, 6.2 Hz, 1H), 4.27–4.15 (m, 3H), 4.13–4.03 (m, 4H), 3.99 (dd, $J = 12.2$, 2.0 Hz, 1H), 2.59 (dd, $J = 14.7$, 3.0 Hz, 1H), 2.39 (dd, $J = 15.1$, 6.0 Hz, 1H), 1.88 (d, $J = 2.8$ Hz, 3H), 1.49 (d, $J = 0.6$ Hz, 3H). $^{13}$C-NMR (1256 MHz, CDCl$_3$): $\delta = 164.4$, 164.3, 151.3, 151.2, 136.3, 135.5, 135.3, 135.2, 134.4, 133.0, 132.3, 130.14, 130.10, 128.0, 124.6, 111.0, 110.9, 90.1, 81.1, 80.7, 78.3, 77.2, 74.1, 72.3, 70.8, 61.8, 50.8, 32.4, 27.0, 19.5, 12.6, 12.1. HRMS (MALDI): calcd for C$_{38}$H$_{44}$N$_4$O$_{11}$NaSi [MNa$^+$] 783.2674, found 783.2668.

### (4a*S*,5*R*,7*R*,7a*R*)-3-[(2*R*,3*R*,4*R*,5*R*)-3,4-Dihydroxy-2-(thymin-1-yl)-tetrahydrofuran-5-yl]methyl-7-hydroxymethyl-5-(thymin-1-yl)-4a,5,7,7a-tetrahydrofuro[3,4-*b*][1,4]dioxine (23)

Under N$_2$ atmosphere, TBAF (1.0 M in THF, 85 µL, 85 µmol) was added to a solution of compound **22** (59 mg, 78 µmol) in anhydrous THF (1.5 mL) at room temperature. The reaction mixture was stirred for 5 h at room temperature. The resulting mixture was concentrated *in vacuo*. The residue (76 mg) was purified by column chromatography (CHCl$_3$/MeOH = 9:1) to give compound **23** (31 mg, 76%) as a white foam. IR $\nu_{max}$ (KBr): 3452, 3376, 1685, 1654, 1637, 1489, 1275 cm$^{-1}$. $^1$H-NMR (400 MHz, DMSO-$d_6$): $\delta = 11.32$ (s, 2H), 7.99 (d, $J = 1.0$ Hz, 1H), 7.43 (d, $J = 1.1$ Hz, 1H), 6.24 (d, $J = 6.5$ Hz, 1H), 6.10 (s, 1H), 5.73 (d, $J = 6.5$ Hz, 1H), 5.53 (t, $J = 5.1$ Hz, 1H), 5.32 (d, $J = 6.0$ Hz, 1H), 5.09 (d, $J = 4.5$ Hz, 1H), 4.47–4.43 (m, 1H), 4.11–4.06 (m, 1H), 4.04–3.99 (m, 2H), 3.88–3.75 (m, 3H), 3.71–3.66 (m, 1H), 2.39–2.28 (m, 2H), 1.80 (d, $J = 0.9$ Hz, 3H), 1.75 (d, $J = 0.8$ Hz, 3H). $^{13}$C-NMR (126 MHz, CD$_3$OD): $\delta = 166.3$, 152.5, 152.4, 138.5, 137.9, 136.6, 125.5, 111.8, 110.8, 91.3, 82.4, 81.6, 79.8, 78.9, 74.3, 74.1, 71.7, 59.8, 49.8, 34.3, 12.5, 12.4. HRMS (MALDI): calcd for C$_{22}$H$_{26}$N$_4$O$_{11}$Na [MNa$^+$] 545.1496, found 545.1490.

### (3*S*,4a*S*,5*R*,7*R*,7a*R*)-3-[(2*R*,3*R*,4*R*,5*R*)-3,4-Dihydroxy-2-(thymin-1-yl)-tetrahydrofuran-5-yl]methyl-5-(thymin-1-yl)-7-hydroxymethylhexahydrofuro[3,4-*b*][1,4]dioxine (24)

20 (w/w) % Pd(OH)$_2$/C (7.4 mg) was added to a solution of compound **23** (37 mg, 71 µmol) in THF (1.5 mL) and MeOH (0.5 mL) at room temperature. After the reaction mixture was stirred for 23 h at room temperature under H$_2$ atmosphere at 1 atm, the mixture was filtered, and the filtrate was concentrated *in vacuo*. The residue (36 mg) was purified by column chromatography (CHCl$_3$/MeOH = 9:1) to give compound **24** (20 mg, 53%) as a white foam. IR $\nu_{max}$ (KBr): 3539, 3445, 3367, 3050, 2926, 1685, 1477, 1422, 1368, 1329, 1271 cm$^{-1}$. $^1$H-NMR (500 MHz, CD$_3$OD): $\delta = 8.28$ (d, $J = 1.2$ Hz, 1H), 7.36 (d, $J = 1.2$ Hz, 1H), 6.11 (d, $J = 6.4$ Hz, 1H), 5.75 (d, $J = 4.5$ Hz, 1H), 4.18 (dd, $J = 6.0$, 4.5 Hz, 1H), 4.08–4.00 (m, 3H), 3.95 (dd, $J = 13.0$, 2.0 Hz, 1H), 3.93–3.86 (m, 3H), 3.77–3.72 (m, 2H), 3.41 (dd, $J = 11.7$, 11.1 Hz, 1H), 1.89 (d, $J = 1.2$ Hz, 3H), 1.83–1.87 (m, 5H). $^{13}$C-NMR (100 MHz, CD$_3$OD): $\delta = 166.5$, 166.3, 152.5, 152.4, 138.5, 138.2, 112.0, 110.4, 91.5, 81.8, 81.6, 81.0, 79.8, 74.9, 74.2, 74.14, 74.12, 72.3, 59.7, 34.2, 12.6, 12.4. HRMS (MALDI): calcd for C$_{22}$H$_{28}$N$_4$O$_{11}$Na [MNa$^+$] 547.1653, found 547.1647.

### (5a*S*,6*R*,8*R*,8a*R*)-(*E*)-4-[[(2*R*,3*R*,4*R*,5*R*)-3-Acethyloxy-4-benzyloxy-2-(thymin-1-yl)-tetrahydrofuran-5-yl]-8-*tert*-butyldiphenylsilyloxymethyl-6-(thymin-1-yl)-5a,6,8,8a-tetrahydro-2*H*-furo[3,4-*b*][1,4]dioxepine (25)

Ac$_2$O (1.6 mL, 16 mmol) was added to a solution of compound **18** (855 mg, 1.1 mmol) in AcOH (15 mL) at room temperature. Then, 10 (v/v)% H$_2$SO$_4$ aq. (58 µL, 0.11 mmol) was added to the solution at 0 °C and the reaction mixture was stirred for 2 h at room temperature. After being neutralized with sat. NaHCO$_3$ aq. at 0 °C, the whole mixture was extracted with AcOEt. The organic phase was washed with sat. NaHCO$_3$ aq. and brine, dried over Na$_2$SO$_4$, and concentrated *in vacuo*. The obtained residue (1.0 g) was dissolved in anhydrous 1,2-dichloroethane (11 mL), and thymine (275 mg, 2.2 mmol) and BSA (1.3 mL, 5.5 mmol) was added under N$_2$ atmosphere at room temperature. The solution was refluxed for 20 min and then cooled to 0 °C. TMSOTf (0.39 mL, 2.2 mmol) was added, and the reaction mixture was refluxed for 14 h. After quenching with sat. NaHCO$_3$ aq. at room temperature, the whole mixture was extracted with AcOEt. The organic phase was washed with water, sat. NaHCO$_3$ aq. and brine, dried over Na$_2$SO$_4$, and concentrated *in vacuo*. The residue (970 mg) was purified by column chromatography (CHCl$_3$/AcOE = 2:1) two times to give compound **25** (365 mg, 37%, 2 steps from compound **18**) as a white foam. IR $\nu_{max}$ (KBr): 3063, 2930, 1687, 1469, 1373, 1269 cm$^{-1}$. $^1$H-NMR (500 MHz, DMSO-$d_6$): $\delta = 11.36$ (s, 1H), 11.34 (s, 1H), 7.66–7.63 (m, 4H), 7.48–7.39 (m, 6H), 7.36 (d, $J = 1.2$ Hz, 1H), 7.34–7.25 (m, 4H), 7.21–7.18 (m, 2H), 6.50 (d, $J = 6.2$ Hz, 1H), 5.89 (d, $J = 5.3$ Hz, 1H), 5.22 (d, $J = 4.2$ Hz, 1H), 5.06 (t, $J = 5.7$ Hz, 1H), 4.67 (t, $J = 7.9$ Hz, 1H), 4.40–4.31 (m, 4H), 4.18 (t, $J = 8.6$ Hz, 1H), 4.15–4.09 (m, 2H), 4.07–3.99 (m, 2H), 3.92 (dd, $J = 12.3$, 4.4 Hz, 1H), 2.02 (s, 3H), 1.86 (d, $J = 1.1$ Hz, 3H), 1.47 (s, 3H), 1.02 (s, 9H). $^{13}$C-NMR (125.7 MHz, CDCl$_3$): $\delta = 170.4$, 163.7, 164.0, 150.9, 150.7, 150.4, 137.3, 137.0, 135.5, 135.3, 134.8, 132.9, 132.4, 130.2, 130.1, 128.4, 128.01, 128.00, 127.98, 127.9, 111.7, 110.9, 109.2, 86.3, 85.5, 84.9, 81.1, 79.5, 78.7, 75.9, 73.7, 73.0, 66.7, 61.4, 27.1, 20.7, 19.6, 12.4, 12.2. HRMS (MALDI): calcd for C$_{47}$H$_{52}$N$_4$O$_{12}$NaSi [MNa$^+$] 915.3249, found 915.3243.

### (5a*S*,6*R*,8*R*,8a*R*)-(*E*)-4-[(2*R*,3*R*,4*R*,5*R*)-4-Benzyloxy-3-hydroxy-2-(thymin-1-yl)-tetrahydrofuran-5-yl]-8-*tert*-butyldiphenylsilyloxymethyl-6-(thymin-1-yl)-5a,6,8,8a-tetrahydro-2*H*-furo[3,4-*b*][1,4]dioxepine (26)

K$_2$CO$_3$ (113 mg, 0.82 mmol) was added to a suspension of compound **25**(365 mg, 0.41 mmol) in MeOH (5 mL) at room temperature. The reaction mixture was stirred for 5 h at room temperature. After quenching with sat. NH$_4$Cl aq. at room temperature, the whole mixture was extracted with AcOEt. The organic phase was washed with sat. NH$_4$Cl aq. and brine, dried over Na$_2$SO$_4$, and concentrated *in vacuo*. The residue (356 mg) was purified by column chromatography (CHCl$_3$/MeOH = 19:1) to give compound **26** (330 mg, 95%) as a white foam. IR $\nu_{max}$ (KBr): 3198, 3066, 2931, 2859, 1690, 1471, 1428, 1403, 1372, 1270 cm$^{-1}$. $^1$H-NMR (500 MHz, CDCl$_3$): $\delta = 9.37$ (s, 1H), 8.93 (s, 1H), 7.69–7.67 (m, 4H), 7.48–7.27 (m, 13H), 6.51 (d, $J = 7.3$ Hz, 1H), 5.77 (d, $J = 6.8$ Hz, 1H), 4.91 (dd, $J = 6.0$, 2.3 Hz, 1H), 4.65 (d, $J = 11.7$ Hz, 1H), 4.52–4.47 (m, 2H), 4.36 (dd, $J = 15.9$, 6.0 Hz, 1H), 4.21 (d, $J = 3.3$ Hz, 1H), 4.16 (d, $J = 2.1$ Hz, 1H), 4.14–4.09 (m, 3H), 3.95–3.91 (m, 3H), 3.22 (d, $J = 8.0$ Hz, 1H), 1.97 (d, $J = 1.1$ Hz, 3H), 1.57 (d, $J = 1.1$ Hz, 3H), 1.14 (s, 9H). $^{13}$C-NMR (126 MHz, CDCl$_3$): $\delta = 164.2$, 164.0, 151.0, 150.7, 150.5, 144.9, 137.1, 136.8, 135.5, 135.3, 135.1, 132.9, 132.4, 130.2, 128.5, 128.1, 128.01, 127.99, 111.6, 110.8, 107.8, 88.3, 85.0, 80.9, 79.3, 78.4, 77.7, 77.2, 72.7, 66.8, 61.2, 27.1, 19.5, 12.3, 12.1. HRMS (MALDI): calcd for C$_{45}$H$_{50}$N$_4$O$_{11}$NaSi [MNa$^+$] 873.3143, found 873.3138.

### (5a$S$,6$R$,8$R$,8a$R$)-($E$)-4-[(2$R$,3$R$,4$R$,5$R$)-4-Benzyloxy-3-hydroxy-2-(thymin-1-yl)-tetrahydrofuran-5-yl]-8-hydroxymethyl-6-(thymin-1-yl)-5a,6,8,8a-tetrahydro-2$H$-furo[3,4-$b$][1,4]dioxepine (27)

Under N$_2$ atmosphere, TBAF (1.0 M in THF, 0.38 mL, 0.38 mmol) was added to a solution of compound **36** (295 mg, 0.35 mmol) in anhydrous THF (4 mL) at room temperature. The reaction mixture was stirred for 6 h at room temperature. After addition of MeOH at room temperature, the resulting mixture was concentrated *in vacuo*. The residue (406 mg) was purified by column chromatography (CHCl$_3$/MeOH = 19:1) to give compound **27** (180 mg, 85%) as a white foam.

IR ν$_{max}$ (KBr): 3456, 1709, 1684, 1658, 1649, 1474, 1459, 1450, 1375, 1266 cm$^{-1}$. $^1$H-NMR (500 MHz, CD$_3$OD): δ = 8.20 (s, 1H), 7.61 (s, 1H), 7.35–7.27 (m, 5H), 6.46 (d, $J$ = 6.8 Hz, 1H), 5.94 (d, $J$ = 7.0 Hz, 1H), 5.07 (dd, $J$ = 5.8, 2.0 Hz, 1H), 4.72 (d, $J$ = 12.1 Hz, 1H), 4.57 (t, $J$ = 7.5 Hz, 1H), 4.50 (d, $J$ = 12.1 Hz, 1H), 4.33 (dd, $J$ = 16.0, 6.0 Hz, 1H), 4.28 (d, $J$ = 3.1 Hz, 1H), 4.18 (dd, $J$ = 12.7, 4.7 Hz, 2H), 4.09 (t, $J$ = 8.7 Hz, 1H), 4.02–3.95 (m, 2H), 3.85 (dd, $J$ = 6.2, 3.4 Hz, 1H), 3.81–3.76 (m, 1H), 1.98 (s, 3H), 1.86 (s, 3H). $^{13}$C-NMR (126 MHz, CD$_3$OD): δ = 166.4, 166.2, 152.9, 152.4, 151.8, 139.0, 138.2, 129.4, 129.2, 128.9, 112.4, 110.8, 109.5, 88.2, 86.5, 82.9, 81.0, 79.9, 78.4, 73.8, 73.4, 67.7, 59.8, 12.6, 12.4. HRMS (MALDI): calcd for C$_{29}$H$_{32}$N$_4$O$_{11}$Na [MNa$^+$] 635.1966, found 635.1960.

### (4$S$,5a$S$,6$R$,8$R$,8a$R$)-4-[(2$R$,3$R$,4$R$,5$R$)-3,4-dihydroxy-2-(thymin-1-yl)-tetrahydrofuran-5-yl]-8-hydroxymethyl-6-(thymin-1-yl)hexahydro-2$H$-furo[3,4-$b$][1,4]dioxepine (28) and (4$R$,5a$S$,6$R$,8$R$,8a$R$)-4-[(2$R$,3$R$,4$R$,5$R$)-3,4-dihydroxy-2-(thymin-1-yl)-tetrahydrofuran-5-yl]-8-hydroxymethyl-6-(thymin-1-yl)hexahydro-2$H$-furo[3,4-$b$][1,4]dioxepine (29)

20 (w/w) % Pd(OH)$_2$/C (90 mg) was added to a solution of compound **27** (180 mg, 0.29 mmol) in THF (6 mL) at room temperature. After the reaction mixture was stirred for 23 h at room temperature under H$_2$ atmosphere at 1 atm, the mixture was filtered, and the filtrate was concentrated *in vacuo*. The residue (140 mg) was purified by column chromatography (CHCl$_3$/MeOH = 9:1) to give a diastereo mixture of compound **28** and **29** (72 mg, 47%). For the NMR-structural analysis, the obtained mixture was purified by reversed-phase HPLC to give compound **28** (26 mg) as a white foam and compound **29** (10 mg) as a white foam, respectively.

Conditions for reversed-phase HPLC-purification
Mobile phase A: H$_2$O
Mobile phase B: MeOH
Linear Gradient of MeOH: 10–70% (30 min)
Column: Nacalai Tesque COSMOSIL 5C$_{18}$-MS-II 5 μm (20 × 250 mm)
Flow rate: 8.0 mL/min
Wavelength of UV for compound detections: 260 nm
Temperature of column oven: 50 °C
Compound **28**: IR ν$_{max}$ (KBr): 3487, 3364, 1691, 1639, 1629, 1560, 1474, 1420, 1272 cm$^{-1}$. $^1$H-NMR (400 MHz, CD$_3$OD): δ = 8.13 (d, $J$ = 1.0 Hz, 1H), 7.68 (d, $J$ = 1.2 Hz, 1H), 6.31 (d, $J$ = 7.1 Hz, 1H), 5.95 (d, $J$ = 6.7 Hz, 1H), 4.77 (dd, $J$ = 8.0, 7.5 Hz, 1H), 4.27–4.18 (m, 2H), 4.16–4.10 (m, 2H), 3.95 (dd, $J$ = 12.8, 2.0 Hz, 1H), 3.91 (dd, $J$ = 6.2, 3.5 Hz, 1H), 3.87 (dt, $J$ = 9.1, 2.0 Hz, 1H), 3.80 (dd, $J$ = 3.5, 1.0 Hz, 1H), 3.77–3.70 (m, 2H), 2.51–2.39 (m, 1H), 1.97 (d, $J$ = 1.1 Hz, 3H), 1.89 (d, $J$ = 1.1 Hz, 3H), 1.87–1.80 (m, 1H). $^{13}$C-NMR (100 MHz, CDCl$_3$): δ = 166.4, 166.3, 153.0, 152.5, 138.5, 138.3, 112.4, 110.6, 89.3, 88.7, 841., 82.9, 81.2, 79.3, 79.2, 73.7, 71.7, 71.3, 60.1, 35.2, 12.6, 12.5. HRMS (MALDI): calcd for C$_{22}$H$_{28}$N$_4$O$_{11}$Na [MNa$^+$] 547.1653, found 547.1647. Compound **29**: IR ν$_{max}$ (KBr): 3677, 3099, 3046, 3017, 3003, 1707, 1542, 1475, 1435, 1268 cm$^{-1}$. $^1$H-NMR (400 MHz, CD$_3$OD): δ = 8.09 (d, $J$ = 1.2 Hz, 1H), 7.43 (d, $J$ = 1.2 Hz, 1H), 6.37 (d, $J$ = 6.4 Hz, 1H), 5.89 (d, $J$ = 7.9 Hz, 1H), 4.49–4.42 (m, 2H), 4.13–4.07 (m, 1H), 4.05–4.01 (m, 2H), 4.00–3.96 (m, 1H), 3.95–3.90 (m, 2H), 3.87–3.82 (m, 2H), 3.73 (dd, $J$ = 12.9, 2.3 Hz, 1H), 2.24–2.13 (m, 1H), 2.02 (ddd, $J$ = 16.2, 5.0, 3.6 Hz, 1H), 1.95 (d, $J$ = 1.1 Hz, 3H), 1.89 (d, $J$ = 1.1 Hz, 3H). $^{13}$C-NMR (100 MHz, CD$_3$OD): δ = 166.4, 166.3, 153.0,

152.5, 138.8, 138.3, 112.2, 110.7, 88.8, 87.7, 84.1, 83.1, 81.1, 80.8, 77.3, 74.2, 70.9, 68.4, 60.0, 34.3, 12.5, 12.4. HRMS (MALDI): calcd for C$_{22}$H$_{28}$N$_4$O$_{11}$Na [MNa$^+$] 547.1653, found 547.1647.

### Conformational analysis of 2′,3′-*trans*-BNAs by ab initio calculation

The theoretical calculation was performed using the Spartan'20 program. The calculation was carried out using density functional theory (DFT) with the basis function 6-31-G* (ωB97XD/6-31-G*) by using the model compounds **23a**, **24a**, **28a**, and **29a**, shown in Fig. 5.

### Synthesis of dT-, rT- and LNA-T-dimers for CD (Circular Dichroism) spectrum measurements

dT-phosphoramidite (Sigma), TBDMS-protected rT-phosphoramidite (Glen Reserach), and LNA-T-phosphoramidite (Fujifilm Wako Pure Chemical) were dissolved in anhydrous MeCN to a final concentration of 0.1 M. The synthesis of dimers was performed on a 10 μmol scale by using an automated DNA synthesizer (Gene Design nS-8 Oligonucleotides Synthesizer) with 0.25 M 5-Ethylthio-1$H$-tetrazole (ETT) in MeCN as an activator. The oligonucleotides synthesized in trityl-on mode were cleaved from the GPG resin by treatment with 28% aqueous NH$_3$ at room temperature for 1.5 h. Removal of NH$_3$ was carried out *in vacuo*. The crude dimers were purified by Sep-Pak® Plus C18 Cartridge (Waters) with the 5′-DMTr group being removed during purification using 1% (v/v) aqueous trifluoroacetic acid. The separated dimers were further purified by reversed-phase HPLC (Waters XBridge® OST C18 Column 2.5 μm, 10 × 50 mm) using 0.1 M TEAA buffer (pH 7.0) and MeCN.

### CD spectrum measurements

Each dimer (3.0–5.0 mg) was dissolved in a H$_2$O/MeOH (1:1) solution of 10 mM sodium phosphate (pH 7.0) to a final concentration of 100 μM. CD spectra were recorded at 20, 40, and 60 °C in the quartz cuvette of 1 cm optical path length. The samples were prepared in the same manner as described in the UV melting experiments. The molar ellipticity was calculated from the equation [θ] = θ/$cl$, where θ indicates the relative intensity, $c$ is the sample concentration, and $l$ means the cell path length in centimeters.

### Data availability

The data underlying this study are available in the published article and the Supporting Information.

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

## Acknowledgements

The funding of the research was partially supported by the Japan Society for the Promotion of Science (JSPS) KAKENHI Grant Numbers JP20K15401, JP23K04930, JP24H00839, and JP24H00840, and by the Japan Agency for

Medical Research and Development (AMED) Grant Numbers JP19am0401003, JP21ae0121022, JP21ae0121023 and JP21ae0121024.

## Author contributions

T.O. designed the experiments. R.N., K.U., and S.M. performed experiments. T.O., R.N., K.U., S.M., and S.O. co-wrote the paper. S.O. supervised the project. All authors have given approval to the final version of the article.

## Competing interests

The authors declare no competing interests.
