## [Transparent Peer Review file · Communications Chemistry]

Synthesis and structural analysis of dinucleotides containing 2',3'-trans-bridged nucleic acids (2',3'-trans-BNAs) with trans-5,6- or 5,7-fused ring skeleton

Corresponding Author: Professor Satoshi Obika

Version 0:

Reviewer comments:

Reviewer #1

(Remarks to the Author)

The manuscript proposed by Pr Obika and Coll. address a very interesting and poorly investigated area in nucleic acid chemistry i.e the design and synthesis of nucleosides constrained in non-canonical conformation with the aim of developing tools to modulate the biological or chemical activities of nucleic acids folded in unusual non-B structures.

In this work the authors designed nucleosides in which four out of six torsional angles of the sugar phosphate backbone can be restrained within six or seven membered ring fused with the ribose moiety (denoted as trans-BNA). This approach is very cleaver but in a synthetic point of view highly challenging. Eventually, three out of four proposed trans-BNA nucleosides have been obtained and their conformation characterized.

Before publication the authors should make some improvements of the manuscript and perhaps furnish some additional results.

In the introduction (page 2, line 10) the authors present the concept of conformational restriction that they have initiated with the BNA in the late nineteen. They use the word "immobilizing sugar moieties" that can be misleading instead of restraining the sugar puckering.

Moreover, in this part they should separate the nucleosides analogues (BNA, LNA, bcNA & tcNA) that have the sugar puckering locked (delta torsional angle fixed, see Lebreton Chem. Rev 2010) from the nucleotides in which it is the sugar/phosphate backbone that is constrained (others torsional angles than delta): CNA and the include nucleotides described in Sekine and col JOC 2000, 65, 3571 & 6515, JOC 1996 and Nielsen, OL 2000, Chem. Com 2002, OBC 2005 and Tetrahedron 2006.

If the design of 2',3'-5,6-trans-BNAs has a good rational with their rigid six membered ring fused with the ribose moiety in order to have defined constrain on torsional angle epsilon to alpha, the design of a floppy seven membered ring in the 2',3'-5,7-trans-BNAs is more surprising. It seems that the latter are obviously more the results of synthetic issues than rational design, could the authors comment about that?

The different steps of synthesis are well described and show that this was not an easy task to achieve. Minor modifications of table 1 with the removal of two useless columns (nb of base eq and Time) and the time column in table 2 also would be nice.

The authors always tried to form the fused cycles using compound 1 or 2 that are made of a nucleoside and a sugar moiety, why didn't they perform this reaction after conversion of the sugar moiety into the corresponding nucleoside especially with 1 that failed to produce the target compound?

Similarly, it turned out that the olefin has been reduced at the dinucleotide level in compound 24 when it failed in its precursor 17, so why didn't they turn back to 17 to built the corresponding dinucleotide isomer of 20 to try to obtain after hydrogenation the other stereoisomer (R) of 24?

Concerning the structural assignment by NMR or calculations means of compounds 23, 24, 28 and 29, the authors properly gave evidence through scrutinization of the coupling constant and NOE effect of the protons involved in the six or seven ring. If the determination of the values of the corresponding torsional angles values is obvious in the case of compound 24 in which the six membered cycle is in chair conformation (there is no need to run computational analysis for this one), the authors should gave more information about the minimized structures of 28a and 29a that feature similar conformation for the seven membered ring which is surprising considering that the methyl group (model of the lower nucleoside) switches from a pseudo axial to a pseudo equatorial position (Figure 5, in which the protons should be removed for clarity). My concern is that the influence of a bigger group such as the nucleoside on the global conformation of the seven membered ring can not be properly represented by the small methyl group and as a consequence provide non accurate estimation of the torsional angles. Moreover, these models avoid the possible stacking interaction between the base that could strongly influence the

overall conformation of the dinucleotides. In that context, circular dichroism experiments would be welcome to get insight of the relative position of the two-thymine bases (see Sekine and Col JOC 1998, 63, 1429).

In the conclusion, it could be interesting to try to compare the obtained non canonical dinucleosides structures with those observed in unusual nucleic acids structures (see Pyle and Col, J. Mol. Biol, 1998, 284,1465) or with previously synthesised nucleotides devoted to mimic local nucleic acid sugar/phosphate backbone distortion (see Catana et al J. Nucleic acid 2012).

The experimental part is relatively well written but special attention should be given to the number of chemical shift mentioned in ^{13}C in correspondance with the number of carbon in the molecule (a chemical shift of -19.4 ppm is denoted for compound 5 whit out any reason). Moreover , a precise assignment of all the signals would be appreciate for the target compounds 23, 24, 28 and 29.

Reviewer #2

(Remarks to the Author)

The present manuscript reports synthesis and conformational characterization of dinucleotides containing 2',3'-trans-bicyclic nucleic acids (BNAs). The long term goal is to use these novel modified dinucleotide analogues to investigate the effect of fixing the conformation of the sugar-phosphate backbone in oligonucleotides. The syntheses were challenging and had been thoroughly explored and optimized to reach the structures that were accessible. The work is well done and the compounds are appropriately characterized. The synthetic part of the manuscript is well written and would be of notable interest to readers of specialized journals, such as, J. Org. Chem. However, the study lacks demonstration of significance and impact on the field of nucleic acid chemistry that would be of interest to the broader readership of Communications Chemistry. The novel nucleoside analogues have not been incorporated in oligonucleotides. This may not be a trivial task because the current synthesis does not provide an easy access to the protected phosphoramidites. More importantly, the interesting differences in backbone conformation have not been evaluated in oligonucleotide model systems. The authors suggest that these unique fixed conformers may find applications in ribozymes and aptamers but these speculations are not supported by any experimental results. The authors conclude that their work developed attractive molecules from the perspective of nucleic acid chemistry; however, the manuscript describes very little possible applications and lacks any studies of the novel compounds in oligonucleotides that would support such applications. In the preset form, the study is an interesting and impressive endeavor in organic chemistry of nucleosides, but lacks the significance and impact required for publication in Communications Chemistry. The manuscript may be reconsidered after major revisions and additional experimental data. The authors should at minimum incorporate and characterize the novel modifications in DNA and RNA model systems, preferably, demonstrating advantageous properties in models having contain bulges or stem-loops that were mentioned in the manuscript as potential structures where the fixed backbones may have unique conformational fits.

Technical comments:

1) The structures of 2',4'-BNA, LNA, bicyclo- and tricyclo-DNAs discussed in introduction will not be obvious to a broader readership. The manuscript readability would be greatly improved by adding these structures to Figure 1.

2) Reference 24 does not compare formacetal modified DNA and RNA. The authors should consider replacing it with the following citation doi/10.1021/ja904926e

3) Page 5, line 11: change "Ohira-Bestman" to "Ohira-Bestmann". In addition, please consider drawing a structure of the reagent in Scheme 2.

Reviewer #3

(Remarks to the Author)

This paper presents the synthesis and structural analysis of a series of "dinucleotides" in which the natural phosphodiester has been replaced by 6- or 7-membered rings fused to the 2',3'-positions of the 5'-terminal nucleoside. This is a well performed synthetic project and a well-written manuscript, which certainly deserves the publications after a few modifications:

1) The introduction is a bit overextended with a range of rather old references. It could be more focused, and for instance, the reference to papers on oligopeptides is out of context. The references for rather old oligonucleotide analogues with modifications or replacements of the phosphodiester could be avoided and/or supplemented by references to newer charged or neutral phosphodiester modifications.

2) Figure 1 shows the five possible dinucleotides targeted, of which all but A were realized. Nevertheless, I wonder why the 2',3'-5,7-trans BNA olefin was not attempted. It would be a product of compound 18, if the Bn group could be removed without reducing the olefin. The compound could be unstable or otherwise not worth the attempt. But it is worth a comment in the manuscript.

3) In Scheme 4, the structure of 26 is not correct, it should be 26 (R1 = TBDPS, R2 = H).

- 4) The yield of 28/29 in Scheme 4 (step e) should be 47% according to text and experimental.
- 5) Obviously, the authors mention that they plan to study the four analyzed dinucleotide analogues in oligonucleotides. However, they need to deal with the 2'-OH group of the 3'-terminal nucleoside, which should be either protected, methylated or fully removed by deoxygenation. I wonder why they didn't from the beginning worked with a 2'-deoxy nucleoside as the starting material. I suggest the way to oligonucleotide is discussed shortly in the discussion part.
- 6) Throughout the experimental, the compound names should be corrected for "franosyl" instead of "furanosyl".

Reviewer 1

We are grateful to reviewer 1 for the useful suggestions that have helped us to improve our paper. As indicated in the responses that follow, we have taken all these comments and suggestions into account in the revised version of our paper.

Q1. In the introduction (page 2, line 10) the authors present the concept of conformational restriction that they have initiated with the BNA in the late nineteen. They use the word "immobilizing sugar moieties" that can be misleading instead of restraining the sugar puckering.

Moreover, in this part they should separate the nucleosides analogues (BNA, LNA, bcNA & tcNA) that have the sugar puckering locked (delta torsional angle fixed, see Lebreton Chem. Rev 2010) from the nucleotides in which it is the sugar/phosphate backbone that is constrained (others torsional angles than delta): CNA and the include nucleotides described in Sekine and col JOC 2000, 65, 3571 & 6515, JOC 1996 and Nielsen, OL 2000, Chem. Com 2002, OBC 2005 and Tetrahedron 2006.

A1. We thank the reviewer's comment. According to the comment, the sentences in the introduction have been revised as follows, to avoid any misunderstanding regarding the puckering of the sugar and phosphate portions,

Revised manuscript (line 10 – line 19 of page 2):

Sugar puckering in nucleic acids occurs such that the N- and S-type conformations are in equilibrium in single-stranded DNA and RNA, whereas the sugar conformation in DNA duplexes is S-type and that in RNA duplexes is N-type.^[25,26] Our group and Prof. Wengel's group have independently developed 2',4'-bridged nucleic acid (2',4'-BNA)^[27,28] or locked nucleic acid (LNA, **Figure 1A**),^[29,30] which has a locked N-type sugar and can form highly stable duplexes with target RNA. To date, artificial nucleotides with bicyclic scaffolds and appropriately fixed conformations of the phosphate backbone, such as constrained nucleic acids (CNAs)^[28–37] and some dinucleotides^[38–44] have been developed. Among these artificial nucleotides, CNAs form stable duplexes with complementary single strands. Additionally, bicyclo DNAs,^[45] and tricyclo DNAs,^[45–48] which have a fixed torsion angle γ , are known to stabilize the duplexes formed with RNA. Moreover, these bicyclo and tricyclo DNAs have a fixed torsion angle δ , which is related to sugar puckering; the torsion angle δ is also fixed for the 2',4'-BNA/LNA described above.

Original (line 11 – line 17 of page 2):

The conformation of the sugar in nucleic acids is in equilibrium between the N- and S-type in single-stranded DNA and RNA, whereas the sugar conformation in DNA duplexes is S-type and that in RNA duplexes is N-type.^[25,26] Our group and Prof. Wengel's group have independently developed 2',4'-bridged nucleic acid (2',4'-BNA)^[27,28] or locked nucleic acid (LNA),^[29,30] which has a locked N-type sugar and can form highly stable duplexes with target RNA. Similarly, artificial nucleotides with bicyclic or tricyclic sugar moieties and appropriately fixed conformations of the phosphate backbone, such as constrained nucleic acids (CNAs),^[31–40] bicyclo-DNAs,^[41] and tricyclo-DNAs,^[42–44] can form stable duplexes with the target RNA.

By the above modifications, references 38–44 have been added.

Additional references in revised manuscript (line 19 of page 30 – line 1 of page 31):

- 38) Seio, K., Wada, T., Sakamoto, K., Yokoyama, S. & Sekine, M. Chemical Synthesis and Conformational Properties of a New Cyclouridylic Acid Having an Ethylene Bridge between the Uracil 5-Position and 5'-Phosphate Group. *J. Org. Chem.* **61**, 1500–1504 (1996).
- 39) Sekine, M., Kurasawa, O., Shohda, K., Seio, K. & Wada, T. Synthesis and Properties of Oligodeoxynucleotides Incorporating a Conformationally Rigid Uridine Unit Having a Cyclic Structure at the 5'-Terminal Site. *J. Org. Chem.* **65**, 3571–3578 (2000).
- 40) Sekine, M., Kurasawa, O., Shohda, K., Seio, K. & Wada, T. Synthesis and Properties of Oligonucleotides Having a Phosphorus Chiral Center by Incorporation of Conformationally Rigid 5'-Cyclouridylic Acid Derivatives. *J. Org. Chem.* **65**, 6515–6524 (2000).
- 41) Sørensen, A. M. & Nielsen, P. Synthesis of Conformationally Restricted Dinucleotides by Ring-Closing Metathesis. *Org. Lett.* **2**, 4217–4219 (2000).
- 42) Børsting, P. & Nielsen, P. Tandem ring-closing metathesis and hydrogenation towards cyclic dinucleotides. *Chem. Commun.* 2140–2141 (2002).

- 43) Børsting, P., Nielsen, K. E. & Nielsen, P. Stabilisation of a nucleic acid three-way junction by an oligonucleotide containing a single 2'-C to 3'-O-phosphate butylene linkage prepared by a tandem RCM-hydrogenation method. *Org. Biomol. Chem.* **3**, 2183–2190 (2005).
- 44) Børsting, P., Christensen, M. S., Steffansen, S. I. & Nielsen, P. Synthesis of dinucleotides with 2'-C to phosphate connections by ring-closing metathesis. *Tetrahedron* **62**, 1139–1149 (2006).

Q2. If the design of 2',3'-5,6-*trans*-BNAs has a good rational with their rigid six membered ring fused with the ribose moiety in order to have defined constrain on torsional angle epsilon to alpha, the design of a floppy seven membered ring in the 2',3'-5,7-*trans*-BNAs is more surprising. It seems that the latter are obviously more the results of synthetic issues than rational design, could the authors comment about that?

A2. We appreciate the reviewer's comments. As the reviewer pointed out, the molecular design of 2',3'-5,7-*trans*-BNAs may not be rational. From a synthetic chemistry point of view, the construction of 5,6- and 5,7-*trans*-fused rings is very difficult, as explained in the introduction. Even in the few successful reactions for the construction of 5,6-*trans*-fused rings, the products are obtained as mixtures of compounds with 5,6- and 5,7-*trans*-fused ring skeletons, similar to our results.

On the other hand, we conducted this study because we believe that structural characterization of both 2',3'-5,6- and 2',3'-5,7-*trans*-BNAs with novel bicyclic sugar moiety structures will lead to a deeper understanding of nucleic acid chemistry. Based on the above, we have revised the background as follows.

Revised manuscript (line 33 of page 2 – line 1 of page 3):

Against this background, we designed 2',3'-5,6-*trans*-BNAs (**A** and **B**, **Figure 1C**) with a *trans*-fused ring skeleton in which a six-membered ring is introduced at the 2'- and 3'-carbons of the sugar moiety. The introduction of this characteristic fused ring skeleton into the sugar moiety not only constrains the sugar conformation to the N-type, but also tightly fixes the torsion angles δ , ϵ , ζ , and α of the backbone structure.

Original (line 31– line 35 of page 2):

Against this background, we designed 2',3'-*trans*-BNAs **A–D** (**Figure 1**) with a *trans*-fused ring skeleton in which a six- or seven-membered ring is introduced at the 2'- and 3'-carbons of the sugar moiety. The introduction of this characteristic fused ring skeleton into the sugar moiety not only constrains the sugar conformation to the N-type, but also tightly fixes the torsion angles δ , ϵ , ζ , α , and β , excluding the torsion angle γ , of the backbone structure.

Revised manuscript (line 3 – line 13 of page 3):

Furthermore, although methods for constructing *cis*-5,6-fused ring skeletons in carbohydrates have been developed,^[69–72] the construction of *trans*-5,6-fused ring skeletons is still difficult. Therefore, the synthesis of 2',3'-5,6-*trans*-BNAs will not only help elucidate how conformational restriction of nucleic acids can improve and control the function of oligonucleotides but also address a fascinating and challenging issue from the perspective of organic chemistry. From the two designed 2',3'-5,6-*trans*-BNAs (**A** and **B**), we have achieved the synthesis of **B**. Surprisingly, 2',3'-5,6-*trans*-BNA^{olefin} (**C**), which has a double bond inside the *trans*-fused ring, was found to be stable in these syntheses. In addition, the syntheses of 2',3'-5,7-*trans*-BNAs (**D** and **E**, **Figure 1C**) were performed to ascertain the structural properties of compounds with 5,7-*trans*-fused rings obtained in the desired fused ring construction. In this study, structural analysis of **B–E** was carried out by NMR measurements and computational methods, and the results were compared with the torsion angles of the phosphate backbone in natural DNA and RNA duplexes.

Original (line 1– line 10 of page 3):

Furthermore, although methods for constructing *cis*-5,6- and *cis*-5,7-fused ring skeletons in carbohydrates have been developed,^[65–68] the construction of *trans*-5,6- or *trans*-5,7-fused ring skeletons is difficult. Therefore, the synthesis of 2',3'-*trans*-BNAs will not only help elucidate how conformational restriction of nucleic acids can improve and control the function of oligonucleotides but also address a fascinating and challenging issue from the perspective of organic chemistry. From the four designed 2',3'-*trans*-BNAs (**A–D**), we have achieved the synthesis of three compounds (**B–D**) (**Figure 1**). Surprisingly, 2',3'-5,6-*trans*-BNA^{olefin} (**E**), which has a double bond inside the *trans*-fused ring, was found to be stable in these syntheses. In this study, structural analysis of **E** and the designed compounds **B–D** was carried out by NMR measurements and computational methods, and the results were compared with the torsion angles of the phosphate backbone in natural DNA and RNA duplexes.

Q3. The different steps of synthesis are well described and show that this was not an easy task to achieve. Minor modifications of table 1 with the removal of two useless columns (nb of base eq and Time) and the time column in table 2 also would be nice.

A3. We thank the reviewer's comment. According to the comment, we have revised Table 1 as follows. On the other hand, the reaction times shown in Table 2 depend on the solvent, so we have not removed the time column in Table 2. We appreciate the reviewer's understanding.

Revised manuscript (line 11 of page 5):

Table 1. Optimization of conditions for coupling reaction between allyl iodide **4** and compound **3**

Entry	Base	Solvent	Temp. (°C)	Yield (%)
1	NaH	DMF	0 to rt	20 (12)
2	NaHMDS	THF	-78 to 0	complex mixture
3	NaH	DMF	-15 to 0	43 (10)
4	NaH	DMF	-15	69 (10)

Original (line 7 of page 5):

Table 1. Optimization of conditions for coupling reaction between allyl iodide **4** and compound **3**

Entry	Base (eq)	Solvent	Temp. (°C)	Time (h)	Yield (%)
1	NaH	DMF	0 to rt	3	20 (12)
2	NaHMDS	THF	-78 to 0	3	complex mixture
3	NaH	DMF	-15 to 0	3	43 (10)
4	NaH	DMF	-15	3	69 (10)

Q4. The authors always tried to form the fused cycles using compound **1** or **2** that are made of a nucleoside and a sugar moiety, why didn't they perform this reaction after conversion of the sugar moiety into the corresponding nucleoside especially with **1** that failed to produce the target compound?

A4. We have also tried several routes to synthesize 2',3'-*trans*-BNAs other than those shown in the original manuscript. These include a synthetic route that reverses the order of glycosylation and cyclization using compound **1** or **2** as suggested by the reviewer. However, only the cyclization reaction using compound **2** shown in Table 2 gave the desired 2',3'-*trans*-fused nucleoside in high yield.

Q5. Similarly, it turned out that the olefin has been reduced at the dinucleotide level in compound **24** when it failed in its precursor **17**, so why didn't they turn back to **17** to build the corresponding dinucleotide isomer of **20** to try to obtain after hydrogenation the other stereoisomer (R) of **24**?

A5. We appreciate the reviewer's question. We also performed the glycosylation from compound **17**. The results showed that most of the substrate **17** was decomposed, probably because vinyl ethers are unstable under acidic conditions for removing acetonide group. The reason is not clear, but it seems that the stability of the vinyl ether increases when the olefin is located inside the fused ring. We appreciate the reviewer's understanding. Related to our earlier response to the reviewer's fourth comment, we have tried many reactions to synthesize 2',3'-*trans*-BNAs. While we would like to present all of them, we would like to focus on the key intramolecular

cyclization reactions in the synthesis of 2',3'-*trans*-BNAs. Therefore, it is difficult to add all the results we have performed, but we have added the following brief comments on the glycosylation of compound **17**.

Revised manuscript (line 9 – line 11 of page 9):

Since the olefin inside the fused ring of compound **23** could be hydrogenated, glycosylation of **17** was attempted to obtain the *R*-isomer of 2',3'-5,6-*trans*-BNA **24**. However, the glycosylation was not successful, probably due to the poor stability of the exocyclic olefin.

- Q6. Concerning the structural assignment by NMR or calculations means of compounds **23**, **24**, **28** and **29**, the authors properly gave evidence through scrutinization of the coupling constant and NOE effect of the protons involved in the six or seven ring. If the determination of the values of the corresponding torsional angles values is obvious in the case of compound **24** in which the six membered cycle is in chair conformation (there is no need to run computational analysis for this one), the authors should give more information about the minimized structures of **28a** and **29a** that feature similar conformation for the seven membered ring which is surprising considering that the methyl group (model of the lower nucleoside) switches from a pseudo axial to a pseudo equatorial position (Figure 5, in which the protons should be removed for clarity). My concern is that the influence of a bigger group such as the nucleoside on the global conformation of the seven membered ring can not be properly represented by the small methyl group and as a consequence provide non accurate estimation of the torsional angles. Moreover, these models avoid the possible stacking interaction between the base that could strongly influence the overall conformation of the dinucleotides. In that context, circular dichroism experiments would be welcome to get insight of the relative position of the two-thymine bases (see Sekine and Col JOC 1998, 63, 1429).
- A6. We appreciate the reviewer's helpful comment. The structures of the four dimers synthesized in this study were analyzed and CD spectra were measured for a total of seven compounds for comparison with the structures of the DNA, RNA, and LNA dimers. The temperature dependence of the CD spectra was also confirmed. The results have been added as Figure 6. In addition, experimental protocols for CD spectrum measurements have been added to the experimental section. (Due to solubility issues with dimers, a 1:1 mixture of water and methanol has been used as a solvent.)

Revised manuscript (line 1 of page 14):

Figure 6. CD spectra of dimers (dT-dimer, rT-dimer, LNA-T-dimer, **23**, **24**, **28**, and **29**). Conditions: 10 mM sodium phosphate (pH 7.0) and 100 μ M dimer in H₂O and MeOH (1:1) at 20, 40, and 60 °C.

Revised manuscript (line 13– line 29 of page 27):

Synthesis of dT-, rT- and LNA-T-dimers for CD (Circular Dichroism) spectrum measurements.

dT-phosphoramidite (Sigma), TBDMS-protected rT-phosphoramidite (Glen Reserach), and LNA-T-phosphoramidite (Fujifilm Wako Pure Chemical) were dissolved in anhydrous MeCN to a final concentration of 0.1 M. The synthesis of dimers was performed on a 10 μ mol scale by using an automated DNA synthesizer (Gene Design nS-8 Oligonucleotides Synthesizer) with 0.25 M 5-Ethylthio-1*H*-tetrazole (ETT) in MeCN as an activator. The oligonucleotides synthesized in trityl-on mode were cleaved from the GPG resin by treatment with 28% aqueous NH₃ at room temperature for 1.5 h. Removal of NH₃ was carried out *in vacuo*. The crude dimers were purified by Sep-Pak® Plus C18 Cartridge (Waters) with the 5'-DMTr group being removed during purification using 1% (v/v) aqueous trifluoroacetic acid. The separated dimers were further purified by reversed-phase HPLC (Waters XBridge® OST C18 Column 2.5 μ m, 10 \times 50 mm) using 0.1 M TEAA buffer (pH 7.0) and MeCN.

CD spectrum measurements.

Each dimer (3.0–5.0 mg) was dissolved in a H₂O/MeOH (1:1) solution of 10 mM sodium phosphate (pH 7.0) to a final concentration of 100 μ M. CD spectra were recorded at 20, 40, and 60 °C in the quartz cuvette of 1 cm optical path length. The samples were prepared in the same manner as described in the UV melting experiments. The molar ellipticity was calculated from the equation $[\theta] = \theta/cl$, where θ indicates the relative intensity, c is the sample concentration, and l means the cell path length in centimeters.

The CD spectra of the four dimers synthesized in this study did not completely match the spectra of DNA, RNA, and LNA. While these results alone do not suggest that the two thymines in our four dimers do not have a stacking interaction at all, the structure of our dimers would not be a typical oligonucleotide helix. Additionally, it seems difficult to discuss whether the introduction of a nucleoside larger than a methyl group into the fused ring skeleton changes the orientation of the 6- or 7-membered rings based on the CD spectral measurements and the results of the conformational calculations.

Although this additional experiment makes it seem difficult to rigorously discuss the structure of our dimers, we have added the following explanation to the discussion.

Revised manuscript (line 13– line 23 of page 12):

Conversely, the *trans*-fused rings in 2',3'-*trans*-BNAs are replaced by a ribonucleoside that is much larger than the methyl group, and the possibility that the conformation of the fused ring skeleton of the dimer may change under the influence of the stacking interaction of the two thymine nucleobases must be considered. Therefore, CD spectra were measured for the four dimers (**23**, **24**, **28**, and **29**), and the obtained spectra were compared with those of typical DNA and RNA dimers (dT- and rT-dimers) and LNA-T-dimers with fixed sugar puckering (**Figure 6**). The CD spectra of the four dimers were not identical to those of dT-, rT-, and LNA-T-dimers. While these results, in isolation, do not suggest that the two thymine nucleobases in 2',3'-*trans*-BNAs (**23**, **24**, **28**, and **29**) do not have any stacking interaction, the structure of our dimer would not be that of a typical oligonucleotide helix. Determination of whether the introduction of a nucleoside into the *trans*-fused ring changes the orientation of the 6- or 7-membered rings is difficult based on the results of CD spectral measurements and conformational calculations.

Q7. In the conclusion, it could be interesting to try to compare the obtained non canonical dinucleosides structures with those observed in unusual nucleic acids structures (see Pyle and Col, J. Mol. Biol, 1998, 284,1465) or with previously synthesised nucleotides devoted to mimic local nucleic acid sugar/phosphate backbone distortion (see Catana et al J. Nucleic acid 2012).

A7. We appreciate the reviewer's comment. We have cited the literature on CNAs (Catana et al J. Nucleic acid 2012) which the reviewer provided and compared them to the structure of our dimers. Since the sugar puckering of our dimers is N-type, so the torsion angle δ values of our dimers were markedly different from those of CNAs. Moreover, the other torsion angles are not exactly the same, and it seems that both CNAs and our dimers are characteristic and attractive molecules.

Considering this comparison, we have added text to the conclusion, and reference 87 has been added.

Revised manuscript (line 2 – line 4 of page 15):

In addition, upon comparison of the 2',3'-*trans*-BNAs developed in this study and previously synthesized nucleotides devoted to mimic local nucleic acid sugar/phosphate backbone distortion,^[87] it was evident that the structures were not identical. Therefore, 2',3'-*trans*-BNAs appear to be a structurally unique nucleic acid analog.

Additional reference in revised manuscript (line 27 – line 29 of page 33):

87) Catana, D. A., Renard, B. L., Maturane, M., Payrastra, C., Tarrat, N. & Escudier, J. M. Dioxaphosphorinane-Constrained Nucleic Acid Dinucleotides as Tools for Structural Tuning of Nucleic Acids. *J. Nucleic Acids* 215876 (2012).

Q8. The experimental part is relatively well written but special attention should be given to the number of chemical shifts mentioned in ¹³C in correspondence with the number of carbon in the molecule (a chemical shift of -19.4 ppm is denoted for compound 5 without any reason). Moreover, a precise assignment of all the signals would be appreciated for the target compounds 23, 24, 28 and 29.

A8. We appreciate the reviewer's comment. The approximately -20 ppm signal seen in the ¹³C-NMR spectrum of compound **5** is assigned to the propargylic carbon of propargylic iodide. Just as alkyl iodides generally have a high-field shift in signal, propargylic iodides seem to have a high-field shift. We were surprised because we did not expect the chemical shift value of -20 ppm for propargylic iodides.

To convey this fact, the experimental section for compound **5** has been revised as follows.

Revised manuscript (line 32 – line 33 of page 19):

^{13}C -NMR (126 MHz, CDCl_3): $\delta = 137.2, 128.5, 128.2, 128.1, 113.3, 103.6, 83.7, 81.4, 81.3, 77.7, 72.5, 68.3, 26.7, 26.3, -19.4$ (propargylic carbon).

Original (line 29– line 30 of page 17):

^{13}C -NMR (126 MHz, CDCl_3): $\delta = 137.2, 128.5, 128.2, 128.1, 113.3, 103.6, 83.7, 81.4, 81.3, 77.7, 72.5, 68.3, 26.7, 26.3, -19.4$.

In addition, according to the comment, ^1H NMR signal assignments for compounds **23**, **24**, **28**, and **29** are shown below the spectra in the Supporting Information.

Revised Supporting Information (page S23):

$\delta = 11.32$ (s, 2H)	3-NH, 3'-NH
7.99 (d, $J = 1.0$ Hz, 1H)	6-H
7.43 (d, $J = 1.1$ Hz, 1H)	6'-H
6.24 (d, $J = 6.5$ Hz, 1H)	1'-H
6.10 (s, 1H)	7'-H
5.73 (d, $J = 6.5$ Hz, 1H)	1'-H
5.53 (t, $J = 5.1$ Hz, 1H)	5'-OH
5.32 (d, $J = 6.0$ Hz, 1H)	2'-OH
5.09 (d, $J = 4.5$ Hz, 1H)	3'-OH
4.47–4.43 (m, 1H)	4'-H
4.11–4.06 (m, 1H)	2'-H
4.04–3.99 (m, 2H)	3'-H, 2'-H,
3.88–3.75 (m, 3H)	5'-H, 3'-H, 4'-H
3.71–3.66 (m, 1H)	5'-H
2.39–2.28 (m, 2H)	5'-H
1.80 (d, $J = 0.9$ Hz, 3H)	5-CH ₃
1.75 (d, $J = 0.8$ Hz, 3H)	5'-CH ₃

Revised Supporting Information (page S25):

$\delta = 8.28$ (d, $J = 1.2$ Hz, 1H)	6-H
7.36 (d, $J = 1.2$ Hz, 1H)	6'-H
6.11 (d, $J = 6.4$ Hz, 1H)	1'-H
5.75 (d, $J = 4.5$ Hz, 1H)	1'-H
4.18 (dd, $J = 6.0, 4.5$ Hz, 1H)	2'-H
4.08–4.00 (m, 3H)	2'-H, 3'-H, 3'-H
3.95 (dd, $J = 13.0, 2.0$ Hz, 1H)	7'-H
3.93–3.86 (m, 3H)	4'-H, 5'-H, 4'-H
3.77–3.72 (m, 2H)	5'-H, 6'-H
3.41 (dd, $J = 11.7, 11.1$ Hz, 1H)	7'-H
1.89 (d, $J = 1.2$ Hz, 3H)	5-CH ₃
1.83–1.87 (m, 5H)	5-CH ₃ , 5'-H

Revised Supporting Information (page S31):

$\delta = 8.13$ (d, $J = 1.0$ Hz, 1H)	6-H
7.68 (d, $J = 1.2$ Hz, 1H)	6'-H
6.31 (d, $J = 7.1$ Hz, 1H)	1'-H
5.95 (d, $J = 6.7$ Hz, 1H)	1'-H
4.77 (dd, $J = 8.0, 7.5$ Hz, 1H)	2'-H
4.27–4.18 (m, 2H)	2'-H, 7'-H
4.16–4.10 (m, 2H)	3'-H, 5'-H
3.95 (dd, $J = 12.8, 2.0$ Hz, 1H)	5'-H
3.91 (dd, $J = 6.2, 3.5$ Hz, 1H)	3'-H
3.87 (dt, $J = 9.1, 2.0$ Hz, 1H)	7'-H
3.80 (dd, $J = 3.5, 1.0$ Hz, 1H)	4'-H
3.77–3.70 (m, 2H)	4'-H, 5'-H
2.51–2.39 (m, 1H)	6'-H
1.97 (d, $J = 1.1$ Hz, 3H)	5-CH ₃
1.89 (d, $J = 1.1$ Hz, 3H)	5-CH ₃
1.87–1.80 (m, 1H)	6'-H

Revised Supporting Information (page S35):

δ = 8.09 (d, J = 1.2 Hz, 1H)	6-H
7.43 (d, J = 1.2 Hz, 1H)	6'-H
6.37 (d, J = 6.4 Hz, 1H)	1'-H
5.89 (d, J = 7.9 Hz, 1H)	1''-H
4.49–4.42 (m, 2H)	2'-H, 2''-H
4.13–4.07 (m, 1H)	7'-H
4.05–4.01 (m, 2H)	3'-H, 5'-H
4.00–3.96 (m, 1H)	5''-H
3.95–3.90 (m, 2H)	3'-H, 7'-H
3.87–3.82 (m, 2H)	4'-H, 4''-H
3.73 (dd, J = 12.9, 2.3 Hz, 1H)	5'-H
2.24–2.13 (m, 1H)	6'-H
2.02 (ddd, J = 16.2, 5.0, 3.6 Hz, 1H)	6''-H
1.95 (d, J = 1.1 Hz, 3H)	5-CH ₃
1.89 (d, J = 1.1 Hz, 3H)	5'-CH ₃

Reviewer 2

We are grateful to reviewer 2 for the comments that have helped us to improve our paper. As indicated in the responses that follow, we have taken the comments into account in the revised version of our paper.

Q9. The novel nucleoside analogues have not been incorporated in oligonucleotides. This may not be a trivial task because the current synthesis does not provide an easy access to the protected phosphoramidites. More importantly, the interesting differences in backbone conformation have not been evaluated in oligonucleotide model systems. The authors suggest that these unique fixed conformers may find applications in ribozymes and aptamers but these speculations are not supported by any experimental results. The authors conclude that their work developed attractive molecules from the perspective of nucleic acid chemistry; however, the manuscript describes very little possible applications and lacks any studies of the novel compounds in oligonucleotides that would support such applications. In the present form, the study is an interesting and impressive endeavor in organic chemistry of nucleosides, but lacks the significance and impact required for publication in Communications Chemistry. The manuscript may be reconsidered after major revisions and additional experimental data. The authors should at minimum incorporate and characterize the novel modifications in DNA and RNA model systems, preferably, demonstrating advantageous properties in models having contain bulges or stem-loops that were mentioned in the manuscript as potential structures where the fixed backbones may have unique conformational fits.

A9. We thank the reviewer's comment. It is reasonable to assume that a series of synthesized derivatives would be valuable to be introduced into oligonucleotides. At present, the introduction of the four synthesized dimers into oligonucleotides has not been achieved, and the applicability of these dimers to aptamers, etc., has not been clearly demonstrated.

On the other hand, it is our understanding that the topics covered in this special issue include not only nucleic acid applications but also the synthetic chemistry of nucleic acids. We are also convinced that the BNAs synthesized in this study are challenging and attractive molecules from the viewpoint of synthetic organic chemistry. We believe that the results of this study deserve to be published in Commun Chem in the context of synthetic organic chemistry of nucleic acids.

Perhaps there is a disagreement between the reviewer and us as to whether the paper is worthy of the journal. We appreciate your understanding of our thoughts.

Q10. The structures of 2',4'-BNA, LNA, bicyclo- and tricyclo-DNAs discussed in introduction will not be obvious to a broader readership. The manuscript readability would be greatly improved by adding these structures to Figure 1.

A10. We appreciate the reviewer's helpful advice. We have added 2',4'-BNA/LNA, bicyclo DNA, and tricyclo DNA to Figure 1 according to the reviewer's comment.

Revised manuscript (line 16 of page 3):

Figure 1. (A) 2',4'-BNA/LNA (B) bicyclo and tricyclo DNA, (C) 2',3'-trans-BNAs

Original (line 13 of page 3):

Figure 1. 2',3'-trans-BNAs

Q11. Reference 24 does not compare formacetal modified DNA and RNA. The authors should consider replacing it with the following citation doi/10.1021/ja904926e

A11. We thank the reviewer for pointing this out. We have confirmed the paper which the reviewer provided, and reference 24 has been replaced with the 2009 JACS paper. Additionally, the reference number has been changed from 24 to 21 in response to other reviewers' remarks.

Revised manuscript (line 6 – line 8 of page 29):

21) Kolarovic, A., Schweizer, E., Greene, E., Gironda, M., Pallan, P. S., Egli, M. & Rozners, E. Interplay of Structure, Hydration and Thermal Stability in Formacetal Modified Oligonucleotides: RNA May Tolerate Nonionic Modifications Better than DNA. *J. Am. Chem. Soc.* **131**, 14932–14937 (2009).

Original (line 21– line 23 of page 26):

24) Rozners, E., Katkevica, D. & Strömberg, R. Oligoribonucleotide Analogues Containing a Mixed Backbone of Phosphodiester and Formacetal Internucleoside Linkages, Together with Vicinal 2'-O-Methyl Groups. *ChemBioChem* **8**, 537–545 (2007).

Q12. Page 5, line 11: change “Ohira–Bestman” to “Ohira–Bestmann”. In addition, please consider drawing a structure of the reagent in Scheme 2.

A12. We appreciate the reviewer's comments. We have revised the text and Scheme 2 as follows.

Revised manuscript (line 2 – line 3 of page 6):

Periodate-mediated cleavage of 1,2-diol **7** and treatment of the generated aldehyde with Ohira–Bestmann reagent gave alkyne **13**.

Original (line 11– line 12 of page 5):

Periodate-mediated cleavage of 1,2-diol **7** and treatment of the generated aldehyde with Ohira–Bestman reagent gave alkyne **13**.

Revised manuscript (line 12 of page 6):

Scheme 2. Synthesis of dimer **2** with internal alkyne

Original (line 8 of page 6):

Scheme 2. Synthesis of dimer **2** with internal alkyne

Reviewer 3

We are grateful to reviewer 3 for the comments that have helped us to improve our paper. As indicated in the responses that follow, we have taken all these comments and suggestions into account in the revised version of our paper.

Q13. The introduction is a bit overextended with a range of rather old references. It could be more focused, and for instance, the reference to papers on oligopeptides is out of context. The references for rather old oligonucleotide analogues with modifications or replacements of the phosphodiester could be avoided and/or supplemented by references to newer charged or neutral phosphodiester modifications.

A13. We appreciate the reviewer's advice. First, in accordance with the comment that oligopeptides are not relevant to this paper, we have revised the text in the Introduction as follows

Revised manuscript (line 24– line 25 of page 1):

As drug targets for conventional pharmaceuticals such as small-molecule drugs are being exhausted, oligonucleotides^[1–5] are attracting attention as a novel drug discovery modality.

Original (line 24– line 25 of page 1):

As drug targets for conventional pharmaceuticals such as small-molecule drugs are being exhausted, oligopeptides^[1–5] and oligonucleotides^[6–10] are attracting attention as novel drug discovery modalities.

In accordance with the comment that citation of oligonucleotide analogues should be avoided and supplemented with references to newer charged or neutral phosphodiester modifications, the text have been revised as follows. With this modification, original references 19–22 have been removed and new references 14–19 have been added.

Revised manuscript (line 3 – line 4 of page 2):

In this context, oligonucleotides that are linked by chemical bonds without phosphorus atoms have also have been explored.^[14–19]

Original (line 2– line 6 of page 2):

In this context, oligonucleotides that are linked by chemical bonds without phosphorus atoms have also been studied, and oligonucleotides with modified backbones such as amide,^[19] formamactal,^[20] and methylene methylimino linkages^[21] have been explored.

Additional references in revised manuscript (line 24 of page 28 – line 3 of page 29):

- 14) Epple, S., Thorpe, C., Baker, Y. R., El-Sagheer, A. H. & Brown, T. Consecutive 5'- and 3'-amide linkages stabilise antisense oligonucleotides and elicit an efficient RNase H response. *Chem. Commun.* **56**, 5496–5499 (2020).
- 15) Vasques, G., Migawa, M. T., Wan, W. B., Low, A., Tanowitz, M., Swayze, E. E. & Seth, P. P. Evaluation of Phosphorus and Non-Phosphorus Neutral Oligonucleotide Backbones for Enhancing Therapeutic Index of Gapmer Antisense Oligonucleotides. *Nucleic Acid Ther.* **32**, 40–50 (2021).
- 16) Baker, Y. R., Thorpe, C., Chen, J., Poller, L. M., Cox, L., Kumar, P., Lim, W. F., Lie1, L., McClorey, G., Epple, S., Singleton, D., McDonough, M. A., Hardwick, J. S., Christensen, K. E., Wood, M. J. A., Hall, J. P., El-Sagheer, A. H. & Brown, T. An LNA-amide modification that enhances the cell uptake and activity of phosphorothioate exon-skipping oligonucleotides. *Nat. Commun.* **13**, 1–11 (2022).
- 17) Kotikam, V. & Rozners, E. Amide-Modified RNA: Using Protein Backbone to Modulate Function of Short Interfering RNAs. *Acc. Chem. Res.* **53**, 1782–1790 (2023).
- 18) Pal, C., Richter M. & Rozners, E. Synthesis and Properties of RNA Modified with Cationic Amine Internucleoside Linkage. *ACS Chem. Biol.* **19**, 249–253 (2024).
- 19) Seio, K., Ohnishi, R., Tachibana, S., Mikagi, H. & Masaki, Y. Synthesis of LNA gapmers that replace a phosphorothioate linkage with a sulfonamide in the gap region, and their ability to form duplexes with complementary RNA targets. *Org. Biomol. Chem.* **23**, 400–409 (2025).

Removed references in original manuscript (line 12 – line 18 of page 26):

- 19) Mesmaeker, A. D., Waldner, A.; Lebreton, J., Hoffmann, P., Fritsch, V., Wolf, R. M. & Freier, S. M. Amides as a New Type of Backbone Modification in Oligonucleotides. *Angew. Chem., Int. Ed. Engl.* **33**, 226–229 (1994).
- 20) Rozners, E. & Strömberg, R. Synthesis and Properties of Oligoribonucleotide Analogs Having Formacetal Internucleoside Linkages. *J. Org. Chem.* **62**, 1846–1850 (1997).
- 21) Yang, X., Han, X., Cross, C., Bare, S., Sanghvi, Y. & Gao, X. NMR Structure of an Antisense DNA·RNA Hybrid Duplex Containing a 3'-CH₂N(CH₃)-O-5' or an MMI Backbone Linker. *Biochemistry*, **38**, 12586–12596 (1999).
- 22) Crooke, S. T. *Annu. Rev. Therapeutic Applications of Oligonucleotides. Pharmacol. Toxicol.* **32**, 329–376 (1992).

Q14. Figure 1 shows the five possible dinucleotides targeted, of which all but A were realized. Nevertheless, I wonder why the 2',3'-5,7-*trans* BNA olefin was not attempted. It would be a product of compound 18, if the Bn group could be removed without reducing the olefin. The compound could be unstable or otherwise not worth the attempt. But it is worth a comment in the manuscript.

A14. We appreciate the reviewer's comments. As the reviewer pointed out, if the olefin inside the 7-membered ring can be stably present in the reaction conditions to remove the benzyl group, we think it would be worthwhile to analyze the structure of the resulting compound (2',3'-5,7-*trans*-BNA olefin). Therefore, we have added the following comment to the conclusion.

Revised manuscript (line 4 – line 7 of page 15):

By contrast, 2',3'-5,7-*trans*-BNA derivatives with an internal olefin in the 7-membered ring (2',3'-5,7-*trans*-BNA^{olefin}) were not obtained when removing the benzyl group of compound 27. If 2',3'-5,7-*trans*-BNA^{olefin} can be obtained by replacing the benzyl group with an appropriate protecting group, a structural analysis should be performed.

Q15. In Scheme 4, the structure of 26 is not correct, it should be 26 (R¹ = TBDPS, R² = H).

A15. We thank the reviewer pointing this out. Compound 26 in Scheme 4 has been modified as follows. (Corrected to R¹=TBDPS, R²=H.)

Revised manuscript (line 11 of page 10):

Original (line 2 of page 10):

Scheme 4. Synthesis of 2',3'-5,7-*trans*-BNAs **28** and **29**

Q16. The yield of **28/29** in Scheme 4 (step e) should be 47% according to text and experimental.

A16. We appreciate the reviewer's comments. The information in the experimental section is correct, and the yield of compound **28/29** listed in Scheme 4 was corrected to 47%.

Revised manuscript (line 13 – line 15 of page 10):

Reagents and conditions: a) Ac₂O, H₂SO₄, AcOH, rt, 2 h; b) thymine, BSA, TMSOTf, DCE, reflux, 14 h, 37% (two steps from **18**); c) K₂CO₃, MeOH, rt, 5 h, 95%; d) TBAF, THF, rt, 6 h, 85%; e) H₂ (1 atm), Pd(OH)₂/C, THF, rt, 23 h, 47% (**28/29** = 3:7).

Original (line 4 – line 6 of page 10):

Reagents and conditions: a) Ac₂O, H₂SO₄, AcOH, rt, 2 h; b) thymine, BSA, TMSOTf, DCE, reflux, 14 h, 37% (two steps from **18**); c) K₂CO₃, MeOH, rt, 5 h, 95%; d) TBAF, THF, rt, 6 h, 85%; e) H₂ (1 atm), Pd(OH)₂/C, THF, rt, 23 h, 37% (**28/29** = 3:7).

Q17. Obviously, the authors mention that they plan to study the four analyzed dinucleotide analogues in oligonucleotides. However, they need to deal with the 2'-OH group of the 3'-terminal nucleoside, which should be either protected, methylated or fully removed by deoxygenation. I wonder why they didn't from the beginning worked with a 2'-deoxy nucleoside as the starting material. I suggest the way to oligonucleotide is discussed shortly in the discussion part.

A17. We appreciate the reviewer's comments. When introducing the series of dimers synthesized in this study into oligonucleotides, the 2'-OH group of the monomer on the 3' side needs to be protected, as the reviewer pointed out. Therefore, we have added the following explanation to the discussion.

Revised manuscript (line 30 – line 33 of page 12):

In the future, we would like to introduce the synthesized 2',3'-*trans*-BNAs into oligonucleotides based on the phosphoramidite chemistry^[84–86] to address this question, despite the challenges of introducing appropriate protective groups selectively to the 2'-OH group in 2',3'-*trans*-BNAs.

Original (line 2 – line 4 of page 12):

In the future, we would like to introduce the synthesized 2',3'-*trans*-BNAs into oligonucleotides based on the phosphoramidite chemistry^[80–82] to address this question.

Q18. Throughout the experimental, the compound names should be corrected for “franosyl” instead of “furanosyl”.

A18. We appreciate the reviewer's comments. We have corrected all misspelled "franosyl" to "furanosyl" as suggested by the reviewer. Since there are many corrections, only their typical modifications are written below.

Revised manuscript (line 28 of page 15):

Ethyl (*E*)-3-[(4*R*)-3-*O*-benzyl-1,2-isopropylidene- α -D-erythrofuranos-4-yl]-2-propenoate (8).

Original (line 25 of page 13):

Ethyl (*E*)-3-[(4*R*)-3-*O*-benzyl-1,2-isopropylidene- α -D-erythrofranos-4-yl]-2-propenoate (8).

Revised manuscript (line 8 of page 16):

(*E*)-3-[(4*R*)-3-*O*-Benzyl-1,2-isopropylidene- α -D-erythrofuranos-4-yl]-2-propen-1-ol (9).

Original (line 5 of page 14):

(*E*)-3-[(4*R*)-3-*O*-Benzyl-1,2-isopropylidene- α -D-erythrofranos-4-yl]-2-propen-1-ol (9).

Revised manuscript (line 23 of page 16):

(4*R*)-(*E*)-3-*O*-Benzyl-4-(3-iodoprop-1-en-1-yl)-1,2-isopropylidene- α -D-erythrofuranose (4).

Original (line 20 of page 14):

(4*R*)-(*E*)-3-*O*-Benzyl-4-(3-iodoprop-1-en-1-yl)-1,2-isopropylidene- α -D-erythrofranosose (4).

Others

The additional CD spectral measurements were performed with the support of two new research collaborators (Mr. Keito Uda and Mr. So Muramoto), so these two collaborators have been added as co-authors.

Revised manuscript (line 3 of page 1):

Takashi Osawa¹, Ryota Nakanishi¹, Keito Uda¹, So Muramoto¹, Satoshi Obika^{1,2,*}

Original (line 3 of page 1):

Takashi Osawa¹, Ryota Nakanishi¹, Satoshi Obika^{1,2,*}

In addition, by the above revisions, the reference numbers have been revised as follows.

Reference numbers in revised manuscript (reference numbers in original manuscript):

Reference 1 (Reference 6); Reference 2 (Reference 7); Reference 3 (Reference 8); Reference 4 (Reference 9); Reference 5 (Reference 10); Reference 6 (Reference 11); Reference 7 (Reference 12); Reference 8 (Reference 13); Reference 9 (Reference 14); Reference 10 (Reference 15); Reference 11 (Reference 16); Reference 12 (Reference 17); Reference 13 (Reference 18); Reference 20 (Reference 23); Reference 21 (Reference 24); Reference 22 (Reference 25); Reference 23 (Reference 26); Reference 24 (Reference 27); Reference 25 (Reference 28); Reference 26 (Reference 29); Reference 27 (Reference 30); Reference 28 (Reference 31); Reference 29 (Reference 32); Reference 30 (Reference 33); Reference 31 (Reference 34); Reference 32 (Reference 35); Reference 33 (Reference 36); Reference 34 (Reference 37); Reference 35 (Reference 38); Reference 36 (Reference 39); Reference 37 (Reference 40); Reference 45 (Reference 41); Reference 46 (Reference 42); Reference 47 (Reference 43); Reference 48 (Reference 44); Reference 49 (Reference 45); Reference 50 (Reference 46); Reference 51 (Reference 47); Reference 52 (Reference 48); Reference 53 (Reference 49); Reference 54 (Reference 50); Reference 55 (Reference 51); Reference 56 (Reference 52); Reference 57 (Reference 53); Reference 58 (Reference 54); Reference 59 (Reference 55); Reference 60 (Reference 56); Reference 61 (Reference 57); Reference 62 (Reference 58); Reference 63 (Reference 59); Reference 64 (Reference 60); Reference 65 (Reference 61); Reference 66 (Reference 62); Reference 67 (Reference 63); Reference 68 (Reference 64); Reference 69 (Reference 65); Reference 70 (Reference 66); Reference 71 (Reference 67); Reference 72 (Reference 68); Reference 73 (Reference 69); Reference 74 (Reference 70); Reference 75 (Reference 71); Reference 76 (Reference 72); Reference 77 (Reference 73); Reference 78 (Reference 74); Reference 79 (Reference 75); Reference 80 (Reference 76); Reference 81 (Reference 77); Reference 82 (Reference 78); Reference 83 (Reference 79); Reference 84 (Reference 80); Reference 85 (Reference 81); Reference 86 (Reference 82).